# Zero-Shot Adaptation of Parameter-Efficient Fine-Tuning in Diffusion Models

**Farzad Farhadzadeh** [1]   **Debasmit Das** [1]   **Shubhankar Borse** [1]   **Fatih Porikli** [1]

## Abstract

We introduce ProLoRA, enabling zero-shot adaptation of parameter-efficient fine-tuning in text-to-image diffusion models. ProLoRA transfers pre-trained low-rank adjustments (e.g., LoRA) from a source to a target model without additional training data. This overcomes the limitations of traditional methods that require retraining when switching base models, often challenging due to data constraints. ProLoRA achieves this via projection of source adjustments into the target model's weight space, leveraging subspace and null space similarities and selectively targeting aligned layers. Evaluations on established text-to-image models demonstrate successful knowledge transfer and comparable performance without retraining.

## 1. Introduction

Recent advances in text-to-image diffusion models like Stable Diffusion XL (Podell et al., 2024) and Imagen (Rombach et al., 2022) have fueled widespread adoption for diverse applications, from photorealistic image creation (Hu et al., 2022; Ruiz et al., 2022; Ye et al., 2023) and artistic rendering (Zhang et al., 2023) to sophisticated image and video editing (Meng et al., 2022; Qi et al., 2023). However, full fine-tuning for each specific task incurs significant storage overhead as model sizes grow. Parameter-Efficient Fine-Tuning (PEFT) methods, such as Low-Rank Adaptation (LoRA) (Hu et al., 2022), mitigate this by learning a small set of parameters representing the weight updates. While effective, LoRA adapters are tightly coupled to their base model, posing a significant challenge when base models are updated or deprecated. Migrating these adapters to new models necessitates retraining, which is often impractical due to resource constraints or the unavailability of the origi-

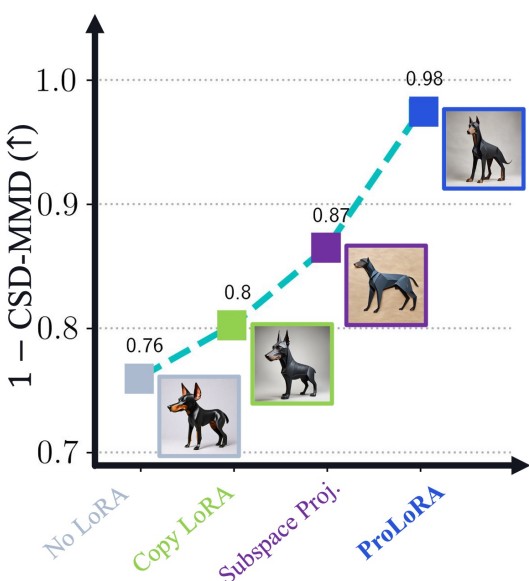

*Figure 1.* Various training-free transfers of LoRA adapter from SDXL to SSD-1B. CSD-MMD is evaluated against LoRA trained on SSD-1B. 'Subspace Proj.' indicates when the null space component is ignored. Higher values on the y-axis indicate better style transfer. Adapter: "Origami", Prompt: "doberman dog".

nal training data.

We introduce ProLoRA, a novel and efficient method for transferring LoRA adapters between diffusion models without retraining or requiring access to the original data. ProLoRA achieves this by meticulously transferring the impact of the source LoRA on both the subspace and null space of the source model's weights to the target model. This preserves the stylistic and functional characteristics of the original adapter. Figure 1 illustrates the effectiveness of ProLoRA in transferring a "Origami" style from a SDXL LoRA to SSD-1B, while quantitative results using our proposed metric, CSD-MMD, demonstrate superior style retention compared to existing baselines.

## 2. Related Work

Parameter-Efficient Fine-Tuning (PEFT) (Xu et al., 2023) has become essential for adapting large pre-trained models to downstream tasks, minimizing computational overhead.

[1]Qualcomm AI Research, San Diego, CA 92121, USA. Qualcomm AI Research is an initiative of Qualcomm Technologies, Inc. Correspondence to: Farzad Farhadzadeh <ffarhadz@qti.qualcomm.com>.

*Proceedings of the 42nd International Conference on Machine Learning*, Vancouver, Canada. PMLR 267, 2025. Copyright 2025 by the author(s).

Various PEFT strategies, including Adapter Modules (Sung et al., 2022), Prompt Tuning (Lester et al., 2021), and Low-Rank Adaptation methods like LoRA (Hu et al., 2022), VeRA (Kopiczko et al., 2023), SVDiff (Han et al., 2023), DoRA (Liu et al., 2024) and FouRA (Borse et al., 2024), aim to achieve efficient adaptation by modifying a limited number of parameters.

Knowledge Distillation (KD) (Hinton, 2015; Gou et al., 2021; Kim & Rush, 2016; Park et al., 2019; Bui Thi Mai & Lampert, 2019) transfers knowledge from a larger teacher model to a smaller student model. Variants like Self-Distillation (Zhang et al., 2019; 2021; Zhang & Sabuncu, 2020) and Weak-to-Strong Distillation (Bang et al., 2021; Kaplun et al., 2022; Wang et al., 2022) offer further refinements. However, KD methods generally require training data, making them unsuitable for data-free scenarios like ours.

Several recent works address the challenge of LoRA transfer. (Wang et al., 2024) employs synthetic data and a small subset of the original dataset for transfer, while (Ran et al., 2023) trains a universal mapper for each target model using a shared dataset subset. In contrast, our proposed method, ProLoRA, offers a training-free, closed-form solution for transferring off-the-shelf LoRAs across different diffusion models.

LoRA-X (Farhadzadeh et al., 2025) shares our goal of training-free LoRA transfer, but with key differences. LoRA-X introduces a specialized LoRA variant that optimizes only singular values, restricting its impact to the weight subspace of the pre-trained model. This limits its flexibility compared to standard LoRA, which can affect both the subspace and nullspace. ProLoRA, on the other hand, provides a general methodology for transferring existing LoRA adapters without modification, preserving their full expressiveness. Furthermore, LoRA-X requires training on the source model before transfer, whereas ProLoRA directly transfers pre-trained LoRAs. Finally, while LoRA-X focuses on style LoRAs, ProLoRA extends to other types like concept LoRA (Ruiz et al., 2022) and LCM-LoRA (Luo et al., 2023b), which pose greater challenges for source model training.

## 3. Motivation

Fine-tuning LoRA adapters ties them to their specific base diffusion model, creating a significant obstacle when migrating to updated, distilled, or pruned versions. Consider transitioning from Stable Diffusion XL (SDXL) (Podell et al., 2024) to a distilled variant like Segmind Stable Diffusion 1B (SSD-1B) (Gupta et al., 2024): directly applying existing SDXL LoRAs to SSD-1B is impossible. Retraining is often impractical due to resource constraints or the unavailability

of the original training data. This inflexibility limits the longevity and broader applicability of LoRA adapters, preventing users from benefiting from advancements in base model architectures.

This work introduces ProLoRA, a novel method for seamlessly transferring LoRA adapters across different diffusion models without retraining or requiring the original training data. ProLoRA leverages the strong correlations observed between layers of different diffusion model versions, particularly in deeper layers where LoRAs exert the greatest influence (Samragh et al., 2023; Frenkel et al., 2024). By precisely mapping the LoRA's impact on both the subspace and nullspace of the source model's weights onto the corresponding spaces of the target model, ProLoRA ensures consistent performance across model architectures. This approach unlocks the full potential of LoRA adaptation, enabling users to easily migrate their customized models to newer and more efficient base models while preserving their carefully tuned functionalities.

## 4. Method

Our method consists of three main parts: **Identifying Module Pairs**: We first need to identify pairs of modules from the source and target models that show high similarity. Since the source and target models might have different numbers of modules, it is crucial to find pairs of modules with the highest similarity. Section 4.1 demonstrates how to measure similarity between each pair. **Decomposing Source LoRA**: Next, we decompose the source LoRA into two components: one that lies in the subspace defined by the source model weights and one in the null space. This decomposition captures the effect of the LoRA on both the subspace and null space. Section 4.1 demonstrates how to decompose the source LoRA. **Transferring Decomposed LoRA**: Finally, we need to transfer the decomposed LoRA to the subspace and null space defined by the weights of the target model. Section 4.3 elaborates on how to perform this transfer.

### 4.1. Subspace Similarity

We begin by applying Singular Value Decomposition (SVD) to $W_s \in \mathbb{R}^{m \times n}$, the source base model weight, and $W_t \in \mathbb{R}^{m \times n}$, the target base model weight, with rank $r_s \leq \min(m, n)$ and $r_t \leq \min(m, n)$, respectively, of a given pair of modules. We obtain $W_s = U_s \Sigma_s V_s^\top$, where $U_s \in \mathbb{R}^{m \times m}$ and $V_s \in \mathbb{R}^{n \times n}$ are left and right singular matrices, respectively, and $\Sigma_s \in \mathbb{R}^{m \times n}$ is a rectangular diagonal matrix of singular values. Similarly, $W_t = U_t \Sigma_t V_t^\top$, where $U_t \in \mathbb{R}^{m \times m}$ and $V_t \in \mathbb{R}^{n \times n}$, are the left and right singular matrices, respectively, and $\Sigma_t \in \mathbb{R}^{m \times n}$ is a rectangular diagonal matrix of singular values.

Following the approach outlined by (Hu et al., 2022;

Farhadzadeh et al., 2025), we utilize

$$\Phi_l(\boldsymbol{W}_s, \boldsymbol{W}_t) = \Psi(\boldsymbol{U}_s, \boldsymbol{U}_t) = \frac{\|\boldsymbol{U}_s^\top \boldsymbol{U}_t\|_F^2}{n} \qquad (1)$$

to measure the column subspace similarity between two matrix weights $\boldsymbol{W}_s$ and $\boldsymbol{W}_t$ of the source and target models. Similarly, we use $\Phi_r(\boldsymbol{W}_s, \boldsymbol{W}_t) = \Psi(\boldsymbol{V}_s, \boldsymbol{V}_t) = \frac{\|\boldsymbol{V}_s^\top \boldsymbol{V}_t\|_F^2}{n}$ to capture row subspace similarity between $\boldsymbol{W}_s$ and $\boldsymbol{W}_t$.

### 4.2. Decomposing Source LoRA

To transfer the adapter $\Delta \boldsymbol{W}_s$, trained on a source model with weights $\boldsymbol{W}_s$, we project $\Delta \boldsymbol{W}_s$ onto the column and row spaces (and their respective null spaces) of $\boldsymbol{W}_s$.

The left singular matrix $\boldsymbol{U}_s$ can be decomposed as $\boldsymbol{U}_s = \begin{bmatrix} \boldsymbol{U}_{s,\|} & \boldsymbol{U}_{s,\perp} \end{bmatrix}$, where $\boldsymbol{U}_{s,\|} \in \mathbb{R}^{m \times r_s}$ contains the orthonormal bases spanning the column subspace of $\boldsymbol{W}_s$, and $\boldsymbol{U}_{s,\perp} \in \mathbb{R}^{m \times (m-r_s)}$ contains the orthonormal bases spanning the null space of $\boldsymbol{W}_s^\top$. Similarly, the right singular matrix $\boldsymbol{V}_s$ can be decomposed as $\boldsymbol{V}_s = \begin{bmatrix} \boldsymbol{V}_{s,\|} & \boldsymbol{V}_{s,\perp} \end{bmatrix}$, where $\boldsymbol{V}_{s,\|} \in \mathbb{R}^{n \times r_s}$ contains the orthonormal bases spanning the row subspace of $\boldsymbol{W}_s$, and $\boldsymbol{V}_{s,\perp} \in \mathbb{R}^{n \times (n-r_s)}$ contains the orthonormal bases spanning the null space of $\boldsymbol{W}_s$. By projecting $\Delta \boldsymbol{W}_s$ to the column (row) and null spaces of $\boldsymbol{W}_s$, we obtain

$$\begin{aligned} \Delta \boldsymbol{W}_s &\approx \boldsymbol{U}_{s,\|} \boldsymbol{U}_{s,\|}^\top \Delta \boldsymbol{W}_s \boldsymbol{V}_{s,\|}^\top \boldsymbol{V}_{s,\|} \\ &\quad + \boldsymbol{U}_{s,\perp} \boldsymbol{U}_{s,\perp}^\top \Delta \boldsymbol{W}_s \boldsymbol{V}_{s,\perp}^\top \boldsymbol{V}_{s,\perp} \\ &= \Delta \boldsymbol{W}_{s,\|} + \Delta \boldsymbol{W}_{s,\perp} \end{aligned} \qquad (2)$$

In the following section, we demonstrate how to transfer each component of the source adapter $\Delta \boldsymbol{W}_s$, i.e., $\Delta \boldsymbol{W}_{s,\|}$ and $\Delta \boldsymbol{W}_{s,\perp}$, to a target model.

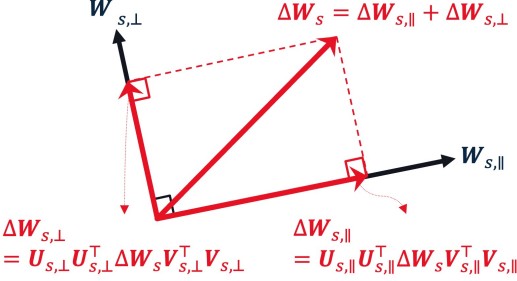

**Figure 2.** Projecting the source adapter into the subspace and null space of the source model weights.

### 4.3. Transferring Decomposed LoRA

Consider $\boldsymbol{W}_s \in \mathbb{R}^{m \times n}$ the source model weight and $\Delta \boldsymbol{W}_s \in \mathbb{R}^{m \times n}$ its corresponding adapter. Our goal is to transfer the adapter to a target model with base model weights $\boldsymbol{W}_t \in \mathbb{R}^{m \times n}$ such that the transferred adapter

$\Delta \boldsymbol{W}_{t \leftarrow s} \in \mathbb{R}^{m \times n}$ has the similar effect on the subspace and null space of $\boldsymbol{W}_t$ as of $\Delta \boldsymbol{W}_s$ on the subspace and null space of $\boldsymbol{W}_s$. To achieve this, we use decomposed $\Delta \boldsymbol{W}_s = \Delta \boldsymbol{W}_{s,\|} + \Delta \boldsymbol{W}_{s,\perp}$ as shown in eq. 2 and project it into the column (row) and null spaces of the base weights of the target model $\boldsymbol{W}_{0,t}$ as follows:

$$\begin{aligned} \Delta \boldsymbol{W}_{t \leftarrow s} &= \boldsymbol{U}_{t,\|} \boldsymbol{U}_{t,\|}^\top \Delta \boldsymbol{W}_{s,\|} \boldsymbol{V}_{t,\|}^\top \boldsymbol{V}_{t,\|} \\ &\quad + \boldsymbol{U}_{t,\perp} \boldsymbol{U}_{t,\perp}^\top \Delta \boldsymbol{W}_{s,\perp} \boldsymbol{V}_{t,\perp}^\top \boldsymbol{V}_{t,\perp} \\ &= \Delta \boldsymbol{W}_{t \leftarrow s,\|} + \Delta \boldsymbol{W}_{t \leftarrow s,\perp} \end{aligned} \qquad (3)$$

where $\boldsymbol{U}_{t,\|}$ and $\boldsymbol{U}_{t,\perp}$ form the right singular matrix $\boldsymbol{U}_t = \begin{bmatrix} \boldsymbol{U}_{t,\|} & \boldsymbol{U}_{t,\perp} \end{bmatrix}$ of $\boldsymbol{W}_t$ the target model weight. Similarly, $\boldsymbol{V}_{t,\|}$ and $\boldsymbol{V}_{t,\perp}$ form the left singular matrix $\boldsymbol{V}_t = \begin{bmatrix} \boldsymbol{U}_{V,\|} & \boldsymbol{U}_{V,\perp} \end{bmatrix}$ of $\boldsymbol{W}_t$.

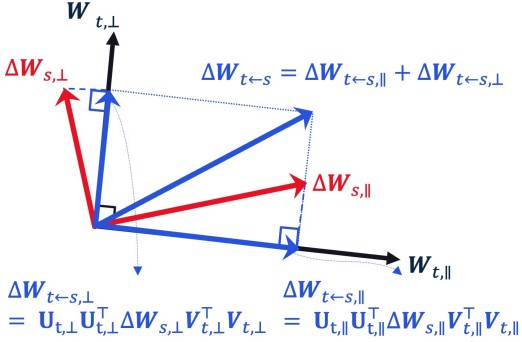

**Figure 3.** Projecting the decomposed source adapter, into the subspace and null space of the target model weights.

When the source and target base model weights have different dimensions (i.e., $m \neq m'$ or $n \neq n'$), we identify a common subspace of equal dimension that maximizes the correlation between the source and target weight subspaces using linear projection, as described in (Farhadzadeh et al., 2025).

LoRA-X (Farhadzadeh et al., 2025) constrains its adapter $\Delta \boldsymbol{W}_s$ to the subspace of $\boldsymbol{W}_s$ (i.e., $\Delta \boldsymbol{W}_s = \Delta \boldsymbol{W}_{s,\|}$). Therefore, the transferred adapter $\Delta \boldsymbol{W}_{t \leftarrow s}$ consists solely of the subspace projection component $\Delta \boldsymbol{W}_{t \leftarrow s,\|}$. In contrast, other adapters like standard LoRA (Hu et al., 2022) are not subject to this constraint, requiring the transfer of both the subspace and nullspace components $\Delta \boldsymbol{W}_{s,\perp}$.

### 4.4. Computation Complexity

While transferring a LoRA adapter requires an initial full SVD computation for both source $\boldsymbol{W}_t \in \mathbb{R}^{m \times n}$ and target $\boldsymbol{W}_t \in \mathbb{R}^{m \times n}$ models ($\mathcal{O}(mn \cdot \min(m, n))$ complexity for each), this cost is amortized over multiple transfers. Subsequent adapter transfers between these pre-processed models are significantly faster than training new LoRAs on the target model, resulting in substantial computational savings.

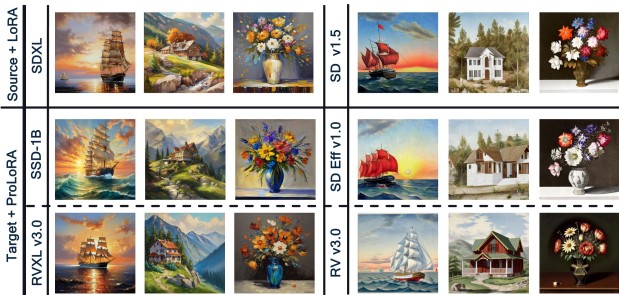

*Figure 4.* Generated samples using the LoRA style adapter trained on SDXL and SD-v1.5 as source models, and the corresponding training-free transferred ProLoRA adapter on SSD-1B and SD Eff-v1.0 as target models. Adapter: "Painting", Prompt: 1) "ship sailing on the sea, sunset" 2) "house on the mountains" 3) "night flowers in vest".

## 5. Experiment

This section describes our experiments to evaluate the effectiveness of ProLoRA in transferring a LoRA from a source to a target diffusion model. We first train the LoRA from scratch for a specific task on both source and target models and then compare the performance of the LoRA trained on the target model with the one transferred from the source model using ProLoRA. We analyze and quantify ProLoRA through text-to-image generation experiments in the following sections, with additional text-generation experiments presented in Appendix E.

### 5.1. Experimental Setup for Text-To-Image Generation

We detail the experimental setup and present our evaluation results, assessing ProLoRA across three types of adapters: (1) style adapters, using datasets with specific styles like origami, (2) concept adapters, with datasets focused on particular subjects, and (3) LCM-LoRA (Luo et al., 2023b) acceleration adapters designed to reduce the number of steps in image generation.

**Datasets:** For style transfer, we are using datasets from public domains, such as *BlueFire*, *Origami Styles*, and *Paintings*. We follow the same setup as described in (Borse et al., 2024; Farhadzadeh et al., 2025). For concept adapter, we use DreamBooth dataset (Ruiz et al., 2022). For transferring acceleration adapter we only use the off-the-shelf LCM-LoRA (Luo et al., 2023b).

**Models:** We employ Stable Diffusion v1.5 (SD-v1.5) (Rombach et al., 2022) and Stable Diffusion XL (SDXL) (Podell et al., 2024) as the source models. SD-v1.5 serves as the source model for target models including Stable Diffusion Efficient v1.0 (SD Eff-v1.0, also used by (Farhadzadeh et al., 2025)), Realistic Vision v3.0 (RV-v3.0). SDXL serves as the source model for target model including Segmind Stable Diffusion 1B (SSD-1B) (Gupta et al., 2024), Realistic

*Table 1.* Comparison of text-to-image generation using LoRAs trained from scratch on target diffusion models versus training-free transfer using ProLoRA. LoRA rank is 32 for all cases.

| Datasets | Base Model | Adapter | HPSv2 (↑) | LPIPS (↑) | CSD-MMD (↓) |
|---|---|---|---|---|---|
| BlueFire (900 images) | RV-v3.0 | LoRA | 0.334 | 0.499 | 0.0061 |
| | | ProLoRA | 0.291 | 0.450 | |
| | SD Eff-v1.0 | LoRA | 0.315 | 0.505 | 0.0025 |
| | | ProLoRA | 0.306 | 0.483 | |
| | RVXL-v3.0 | LoRA | 0.321 | 0.461 | 0.0013 |
| | | ProLoRA | 0.308 | 0.442 | |
| | SSD-1B | LoRA | 0.323 | 0.448 | 0.0207 |
| | | ProLoRA | 0.318 | 0.413 | |
| Paintings (630 images) | RV-v3.0 | LoRA | 0.303 | 0.453 | 0.0034 |
| | | ProLoRA | 0.298 | 0.397 | |
| | SD Eff-v1.0 | LoRA | 0.287 | 0.451 | 0.0026 |
| | | ProLoRA | 0.276 | 0.445 | |
| | RVXL-v3.0 | LoRA | 0.326 | 0.438 | 0.0016 |
| | | ProLoRA | 0.305 | 0.412 | |
| | SSD-1B | LoRA | 0.328 | 0.436 | 0.0134 |
| | | ProLoRA | 0.318 | 0.433 | |
| Origami (900 images) | RV-v3.0 | LoRA | 0.269 | 0.454 | 0.0039 |
| | | ProLoRA | 0.276 | 0.410 | |
| | SD Eff-v1.0 | LoRA | 0.253 | 0.414 | 0.0025 |
| | | ProLoRA | 0.257 | 0.441 | |
| | SSD-1B | LoRA | 0.244 | 0.351 | 0.0245 |
| | | ProLoRA | 0.2560 | 0.3434 | |

Vision XL v3.0 (RVXL-v3.0), SDXL-LCM and SSD-1B-LCM (Luo et al., 2023a), as well as their LCM-LoRA counterparts (Luo et al., 2023b).

**Metrics:** To quantify the quality of images generated by LoRA and its transferred version using ProLoRA, we report the DINOv2 (Oquab et al., 2024), HPSv2.1 (Wu et al., 2023), and LPIPS (Zhang et al., 2018) diversity scores, as well as CSD-MMD. DINOv2 assesses image similarity based on embedded representations. The HPSv2 metric evaluates image quality and alignment with the prompt/style. The LPIPS diversity score captures the diversity among all possible pairs of generated images across different seeds. Additionally, we use MMD (Smola et al., 2006) on CSD (Somepalli et al., 2024) embedded representations to demonstrate how LoRA style is transferred. Specifically, for two image sets, we obtain the CSD descriptors using the ViT backbone and compute the MMD between the features of the two image sets. This metric provides an indication of whether the two image sets have a similar style, with a lower score being better.

### 5.2. Performance of LoRA Transfer

To enable training-free adapter transfer between source and target base models, we first identify correlated modules using equation 1. Following (Farhadzadeh et al., 2025), a threshold of 0.8 is applied to select the most relevant modules. The source LoRA is then projected onto its corresponding target module using equation 2.

### 5.2.1. STYLE LORA

Table 1 compares the performance of LoRA style adapters trained directly on various models (using BlueFire, Painting, and Origami datasets) against ProLoRA, our training-free transfer method. The similarity in HPSv2 and LPIPS scores demonstrates ProLoRA's effectiveness, achieving comparable performance to training from scratch. High DINOv2 scores indicate strong correlation between the generated samples, while low CSD-MMD confirms successful style transfer.

Figure 4 showcases generated samples based on the Painting dataset. The top row displays samples from source models (SDXL and SD-v1.5) using directly trained LoRAs. The following rows present samples from target models (SSD-1B, SD Eff-v1.0, RVXL-v3.0, and RV-v3.0) using transferred ProLoRAs. Qualitative visualizations for BlueFire and Origami are in Appendix A.

Beyond models with identical sampling steps, ProLoRA effectively transfers across models with different sampling configurations. Table 2 presents ProLoRA performance when transferring style LoRAs between standard diffusion models (SDXL, SSD-1B) and their 4-step LCM counterparts (SDXL-LCM, SSD-LCM). Notably, DINOv2 scores remain consistent for "within-model" transfers (e.g., SDXL to SDXL-LCM), suggesting ProLoRA preserves the distributional relationship. However, larger DINOv2 differences are observed for cross-model transfers (e.g., SDXL to SSD-LCM), likely due to architectural differences and incomplete LoRA transfer.

Finally, Figure 5 presents samples generated by LCM models (4-step sampling) using ProLoRAs transferred from standard diffusion models (20-step sampling).

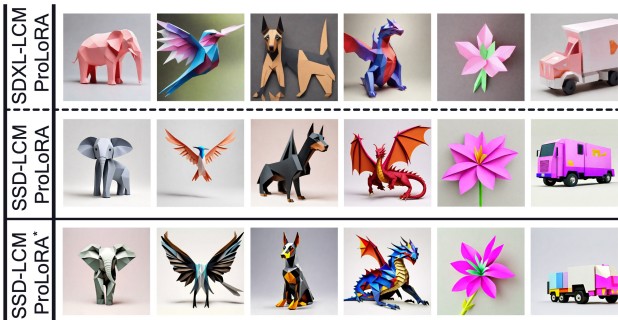

*Figure 5.* Training-free LoRA transfer using ProLoRA. Top: SDXL LoRAs transferred to SDXL-LCM. Middle: SDXL LoRAs transferred to SSD-LCM. Bottom: SSD-1B LoRA transferred to SSD-LCM. All samples generated in 4 steps. Adapter: "Origami". Prompts: 1) "elephant" 2) "bird with spread wings" 3) "doberman dog" 4) "dragon" 5) "flower" 6) "truck".

### 5.2.2. CONCEPT LORA

We also investigated the effect of ProLoRA on concept-specific LoRAs. For this, source model (SDXL) adapters were fine-tuned on each Dreambooth dataset concept using both denoising and prior preservation losses. Each concept adapter was then transferred to the target model (SSD-1B) using ProLoRA. Table 3 presents the transfer results, evaluated with DINOv2, CLIP-I, and CLIP-T metrics. These metrics clearly indicate that ProLoRA achieves quantitative performance close to training from scratch, significantly outperforming direct LoRA copying from source to target and the "No LoRA" baseline, which yields poor results.

*Table 2.* Evaluation of training-free transferred style LoRA from diffusion source models (SDXL, SSD-1B) to LCM versions (SDXL-LCM, SSD-LCM 4 steps) using the Origami dataset.

| Method | HPSv2 (↑) | LPIPS (↑) | DINOv2 (↑) |
|---|---|---|---|
| SDXL LoRA | 0.244 | 0.346 | 0.920 |
| SDXL-LCM ProLoRA | 0.246 | 0.307 | |
| SDXL w/o LoRA | 0.259 | 0.358 | 0.910 |
| SDXL-LCM w/o LoRA | 0.2580 | 0.3753 | |
| SSD LoRA | 0.244 | 0.351 | 0.916 |
| SSD-LCM ProLoRA | 0.247 | 0.346 | |
| SSD w/o LoRA | 0.271 | 0.297 | 0.925 |
| SSD-LCM w/o LoRA | 0.259 | 0.257 | |
| SDXL LoRA | 0.244 | 0.346 | 0.928 |
| SDXL to SSD-LCM ProLoRA | 0.245 | 0.328 | |
| SDXL w/o LoRA | 0.259 | 0.358 | 0.906 |
| SSD-LCM w/o LoRA | 0.259 | 0.257 | |

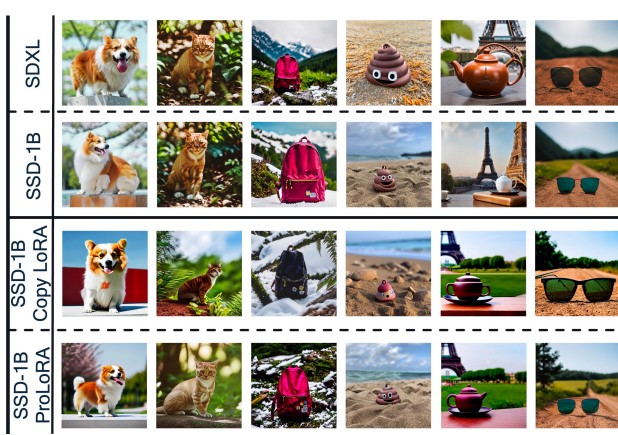

*Figure 6.* Comparison of DreamBooth-trained and transferred LoRAs. Rows 1-2: SDXL and SSD-1B with concept LoRAs trained using DreamBooth. Rows 3-4: SSD-1B with LoRAs transferred from SDXL using copying with subspace similarity matching and ProLoRA. Prompt: 1) "a cube shaped sks dog" 2) "a sks cat in the jungle" 3) "a sks backpack in the snow" 4) "a sks toy in a beach" 5) "a sks toy with the Eiffel tower in the background" 6) "a sks glasses on top of a dirt road".

*Table 3.* Evaluating ProLoRA for training-free concept LoRA transfer from SDXL to SSD-1B on the Dreambooth dataset. Performance is compared to direct LoRA copy, No LoRA, and LoRA fine-tuned from scratch on SSD-1B.

| Method | CLIP-T ($\uparrow$) | CLIP-I ($\uparrow$) | DINOv2 ($\uparrow$) |
|---|---|---|---|
| No LoRA | 0.251 | 0.521 | 0.352 |
| LoRA | 0.294 | 0.745 | 0.539 |
| Copy LoRA | 0.300 | 0.719 | 0.475 |
| ProLoRA | 0.287 | 0.737 | 0.501 |

### 5.2.3. LCM LoRA

Table 4 compares the performance of LCM-LoRA using checkpoints (Luo et al., 2023b) and the training-free transferred Pro-LCM-LoRA. The similar HPSv2 and LPIPS scores indicate that Pro-LCM-LoRA performs comparably to the trained LCM-LoRA, demonstrating its effectiveness. High DINOv2 scores suggest strong correlation in generated samples for both methods. Notably, LCM-LoRA applies the LoRA adapter to both linear and convolutional layers, showing ProLoRA's capability to handle Conv layers as well. Figure 7 shows several samples generated by LCM-LoRA with 4 steps. The first row displays samples generated by the source models SDXL using LCM-LoRA, while the second and third rows show samples generated by training-free transferred Pro-LCM-LoRA and copy-LCM-LoRA (simply copying LCM-LoRA from the source to the target on modules with high subspace similarity) to the target models SSD-1B.

*Table 4.* Evaluation of training-free transferred LCM-LoRA from SDXL to SSD-1B. Results are shown using the evaluation prompt of the Bluefire dataset after removing the trigger word versus LCM-LoRA trained on SDXL from scratch using BlueFire dataset.

| Method | HPSv2 ($\uparrow$) | LPIPS ($\uparrow$) | DINOv2 ($\uparrow$) |
|---|---|---|---|
| LCM-LoRA | 0.329 | 0.494 | —— |
| Copy LCM-LoRA | 0.276 | 0.483 | 0.885 |
| Pro-LCM-LoRA | 0.315 | 0.497 | 0.944 |

### 5.3. Ablation Studies

#### 5.3.1. IMPACT OF NULL SPACE AND SUBSPACE

We analyze the role of the nullspace in LoRA transfer by examining source $\|\Delta W_s\|$ and transferred $\|\Delta W_{t \leftarrow s}\|$ LoRA norms. Figure 8a-c visualizes the norm relationship between LoRAs trained on SD-v1.5 (source) and transferred to SD Eff-v1.0 (target) for the Origami dataset. The strong correlation in the overall norm (Figure 8a) indicates near-perfect transfer, attributed to the high subspace similarity demonstrated by (Farhadzadeh et al., 2025). Decomposing the norm into subspace and nullspace components (Figure 8b-c) reveals high correlation in both. Notably, the

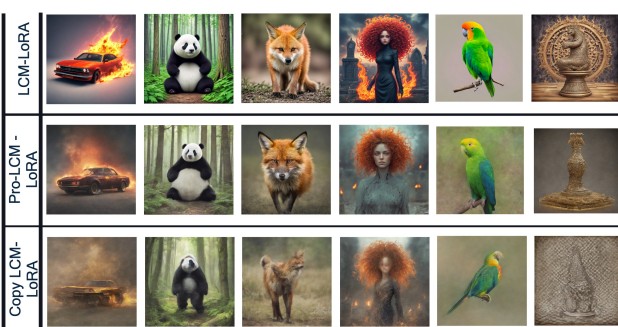

*Figure 7.* Comparison of sample generation using LCM-LoRA in SSD-1B (4 steps). Row 1: LCM-LoRA trained directly on SSD-1B. Row 2: LCM-LoRA transferred from SDXL using ProLoRA. Row 3: LCM-LoRA weights copied from SDXL with subspace similarity matching. Prompt: 1) "blazing fiery car, lightning" 2) "panda in the woods" 3) "ferocious fox, high resolution" 4) "flaming medussa in the graveyard, curly hair" 5) "beautiful parrot, long beak" 6) "blazing chess rooke, intricate work".

nullspace norms range (0-5) exceeds the subspace norms range (0-3), underscoring the nullspace's significance. This range difference originates from LoRAs applied to fully connected layers, which are crucial for capturing complex relationships. Figure 8d demonstrates the high correlation between the norms of a transferred LoRA and a LoRA trained from scratch on SD Eff-v1.0, explaining the effectiveness of training-free transfer. Appendix B presents the same analysis for SDXL (source model) and SSD-1B (target model).

To further investigate the nullspace's impact, we compare ProLoRA, our proposed transfer method, to variants that ablate different components. We trained a LoRA on SDXL (source) with the Origami dataset and transferred it to SSD-1B (target). "ProLoRA w/o NS" ignores the nullspace projection (second term in equation 3), considering only the subspace. Table 5 shows a significantly higher CSD-MMD for ProLoRA w/o NS compared to ProLoRA, indicating deficient style transfer. Conversely, "ProLoRA w/o SS," which only projects into the nullspace, performs similarly to having no LoRA, confirming the subspace's critical role. Finally, applying ProLoRA only to modules with non-square weight matrices ("Where NS Proj." in Table 5), where a nullspace exists, also results in high CSD-MMD, demonstrating the importance of leveraging subspace similarity wherever it is present. Qualitative visualizations are in Appendix B.

#### 5.3.2. COPY LORA

We evaluate ProLoRA against directly copying LoRAs from the source (SDXL) to the target (SSD-1B) model. While copying still can involve identifying the closest module based on subspace similarity, it omits the crucial subspace and nullspace projections employed by ProLoRA. Although not explicitly formulated like equation 3, Copy LoRA can

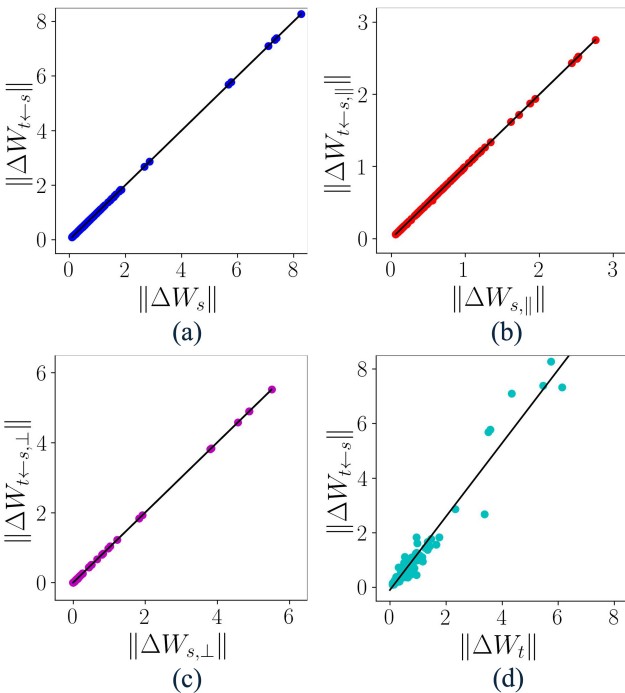

*Figure 8.* Correlation between SD-v1.5 (source) LoRA and transferred LoRAs (ProLoRA) to SD Eff-v1.0 (target): (a) full LoRA norms, (b) subspace components, and (c) nullspace components, and (d) correlation between the norms of a transferred LoRA and a LoRA trained from scratch on SD Eff-v1.0.

*Table 5.* Effect of null space projection in ProLoRA by comparing samples genrated by SSD-1B using style LoRA trained from scratch using the Origami dataset against various ProLoRA variations transferred from SDXL.

| Method | HPSv2 (↑) | LPIPS (↑) | CSD-MMD(↓) |
|---|---|---|---|
| LoRA | 0.244 | 0.351 | —– |
| ProLoRA | 0.256 | 0.343 | **0.0245** |
| ProLoRA w/o NS | 0.270 | 0.313 | 0.1344 |
| ProLoRA w/o SS | 0.265 | 0.312 | 0.2120 |
| Where NS Proj. | 0.265 | 0.331 | 0.1824 |
| Copy w/ SS LoRA | 0.269 | 0.316 | 0.1663 |
| Copy w/o SS LoRA | 0.264 | 0.309 | 0.1968 |
| No LoRA | 0.271 | 0.297 | 0.2394 |

be represented similarly:

$$\Delta \boldsymbol{W}_{t \overset{C}{\leftarrow} s} = \boldsymbol{U}_{t,\|}\boldsymbol{U}_{t,\|}^{\top}\Delta \boldsymbol{W}_s \boldsymbol{V}_{t,\|}^{\top}\boldsymbol{V}_{t,\|} + \boldsymbol{U}_{t,\perp}\boldsymbol{U}_{t,\perp}^{\top}\Delta \boldsymbol{W}_s \boldsymbol{V}_{t,\perp}^{\top}\boldsymbol{V}_{t,\perp}$$

It effectively performs a direct transfer without the alignment terms. For instance, comparing the expansion of Copy LoRA to equation 3 reveals the absence of alignment terms such as $\boldsymbol{U}_{t,\|}\boldsymbol{U}_{t,\|}^{\top}\boldsymbol{U}_{s,\|}\boldsymbol{U}_{s,\|}^{\top}$, which project the source subspace onto the target subspace. While utilizing subspace similarity for module pairing mitigates misalignment to some extent, we demonstrate that these explicit alignment operations within ProLoRA significantly impact transfer performance.

This comparison includes style, concept, and LCM-LoRAs. For style LoRAs (Table 5), copying with ("Copy w/ SS LoRA") and without ("Copy w/o SS LoRA") subspace similarity yields comparable HPSv2 and LPIPS scores to Pro-LoRA, but significantly higher CSD-MMD reveals inferior style transfer. Qualitative visualizations are in Appendix B.

Similar results are observed for concept LoRAs (Table 3). Despite similar quantitative metrics, potentially due to the subspace similarity criterion used during copying, the generated samples (Figure 6, rows 3 vs. 4) reveal inconsistencies in object transfer with copied LoRAs.

Finally, for LCM-LoRAs (Table 4), copying proves ineffective, particularly as evidenced by lower DINOv2 scores. Figure 7 (row 3) further highlights the detrimental impact of copied LCM-LoRAs on image generation quality.

### 5.3.3. LoRA Rank Effect on Transferrability

This analysis investigates the impact of LoRA rank on Pro-LoRA's performance when transferring adapters from SD-v1.5 to SD Eff-v1.0. As Table 6 illustrates, the CSD-MMD between samples generated by SD Eff-v1.0 using natively trained LoRAs and those using ProLoRA-transferred Lo-RAs (originally trained on SD-v1.5) generally increases as the adapter rank decreases, with the exception of the BlueFire dataset. This trend likely arises from the inherent information loss during ProLoRA's cross-model transfer process. ProLoRA projects the LoRA adapter onto the subspace and null space of the target model (SD Eff-v1.0), but these subspaces are not perfectly aligned with those of the source model (SD-v1.5). Consequently, lower ranks provide less capacity for information transfer, exacerbating this misalignment and increasing susceptibility to transfer loss.

*Table 6.* Evaluating the Impact of LoRA Rank on Transferring LoRA from SD-v1.5 to SD Eff-v1.0.

| Dataset | Adapter | Rank | HPSv2 (↑) | LPIPS (↑) | CSD-MMD (↓) |
|---|---|---|---|---|---|
| | LoRA | 32 | 0.315 | 0.505 | |
| | ProLoRA | | 0.306 | 0.483 | 0.0025 |
| BlueFire | LoRA | 16 | 0.265 | 0.525 | |
| | ProLoRA | | 0.312 | 0.506 | 0.0024 |
| | LoRA | 1 | 0.265 | 0.531 | |
| | ProLoRA | | 0.126 | 0.307 | 0.0017 |
| | LoRA | 32 | 0.287 | 0.451 | |
| | ProLoRA | | 0.276 | 0.445 | 0.0026 |
| Paintings | LoRA | 16 | 0.296 | 0.440 | |
| | ProLoRA | | 0.279 | 0.457 | 0.0035 |
| | LoRA | 1 | 0.295 | 0.469 | |
| | ProLoRA | | 0.282 | 0.448 | 0.0042 |
| | LoRA | 32 | 0.253 | 0.414 | |
| | ProLoRA | | 0.257 | 0.441 | 0.0025 |
| Origami | LoRA | 16 | 0.261 | 0.460 | |
| | ProLoRA | | 0.254 | 0.438 | 0.0038 |
| | LoRA | 1 | 0.255 | 0.480 | |
| | ProLoRA | | 0.259 | 0.492 | 0.0047 |

### 5.3.4. SENSITIVITY TO SUBSPACE SIMILARITY THRESHOLD

The initial subspace similarity threshold of 0.8 was selected based on empirical analysis. To assess ProLoRA's sensitivity to this hyperparameter, we experimented with thresholds of 0.9 and 1.0 during LoRA transfer from SDv1.5 to Eff v1.0. As shown in Table 7, these initial results indicate that ProLoRA exhibits relative robustness to variations in this threshold.

*Table 7.* Performance sensitivity of ProLoRA to the subspace similarity threshold during LoRA transfer from SDv1.5 to Eff v1.0 on the BlueFire dataset.

| Threshold | CSD-MMD (↓) |
|-----------|-------------|
| 0.8 | 0.0025 |
| 0.9 | 0.0031 |
| 1.0 | 0.0082 |

### 5.3.5. ITERATIVE TRANSFER ACROSS CHAIN OF MODELS

To assess iterative transfer, we compared chained transfers (SD1.5 → RV3 → EffNet v1.0) against direct transfer (SD1.5 → EffNet v1.0). As shown in Table 8, the results indicate that iterative transfer degrades performance, particularly on the Origami dataset, potentially due to error accumulation.

*Table 8.* Performance sensitivity of ProLoRA to the chain of iterative transfer.

| Dataset | Chain | CSD-MMD (↓) |
|---------|-------|-------------|
| Painting | SD-1.5 → Eff v1.0 | 0.0026 |
| | SD-1.5 → RV-3 → Eff v1.0 | 0.0027 |
| Origami | SD-1.5 → Eff v1.0 | 0.0025 |
| | SD-1.5 → RV-3 → Eff v1.0 | 0.0045 |
| BlueFire | SD-1.5 → Eff v1.0 | 0.0025 |
| | SD-1.5 → RV-3 → Eff v1.0 | 0.0025 |

### 5.4. Transferring DoRA Adapters with ProLoRA

This section demonstrates the effectiveness of ProLoRA for transferring DoRAs (Liu et al., 2024) from a source model (SDXL) to a target model (SSD-1B). We apply our projection method to both the up and down matrices of the DoRA adapter. Quantitative results for the paintings and origami datasets are presented in Table 9, showing that transferring DoRAs via ProLoRA yields performance comparable to training DoRAs from scratch on the target model (SSD-1B). Qualitative visualization results are shown in Appendix C.

*Table 9.* ProLoRA performance on transferring style DoRA from SDXL to SSD-1B. DoRA rank is 8.

| Dataset | Adapter | HPSv2 (↑) | LPIPS (↑) | CSD-MMD (↓) |
|---------|---------|-----------|-----------|-------------|
| Paintings | DoRA | 0.304 | 0.462 | 0.0145 |
| | ProLoRA | 0.307 | 0.472 | |
| Origami | DoRA | 0.249 | 0.341 | 0.0101 |
| | ProLoRA | 0.234 | 0.315 | |

### 5.5. Transferring FouRA Adapters with ProLoRA

This section explores the application of ProLoRA to cross-model transfer of Fourier Low-Rank Adapters (FouRAs) (Borse et al., 2024), specifically from SD-v1.5 to RV3.0. By projecting both the up and down matrices of the FouRA adapter, we facilitate adaptation to the target model. Table 10 presents quantitative results on the paintings dataset, demonstrating the efficacy of this approach, with transferred FouRA performance rivaling that of training from scratch on RV3.0. Qualitative visualization results are shown in Appendix D.

*Table 10.* ProLoRA performance on transferring style FouRA from SD-v1.5 to RV3.0. FouRA rank is 64.

| Dataset | Adapter | HPSv2 (↑) | LPIPS (↑) | CSD-MMD (↓) |
|---------|---------|-----------|-----------|-------------|
| Paintings | FouRA | 0.303 | 0.469 | 0.0023 |
| | ProLoRA | 0.307 | 0.464 | |

### 5.6. Comparison with X-adapter

We compare the performance of our training-free LoRA transfer method, ProLoRA, with X-Adapter (Ran et al., 2023), which utilizes plug-and-play modules trained on the target model. Table 11 presents this comparison. ProLoRA denotes our training-free transfer from SSD-1B to SDXL. X-Adapter refers to their transfer method using modules trained for adaptation from SD-v1.5 to SDXL. LoRA represents a LoRA adapter trained from scratch on the BlueFire dataset using the target model (SDXL). The results show that HPSv2 and LPIPS have similar performance changes from the trained baseline. However, ProLoRA achieves a higher DINOv2 score due to its transfer from a related source, SSD-1B. Additionally, X-adapter has longer inference times because it processes through the base model, transferred model, and adapter.

*Table 11.* Evaluation of LoRA trained from scratch on SDXL versus training-free transferred ProLoRA from SSD-1B into SDXL and X-adapter from SD-v1.5 to SDXL using BlueFire dataset. Wall clock inference time is measured on A100 GPU.

| Adapter | HPSv2 (↑) | LPIPS (↑) | DINOv2 (↑) | Time (↓) |
|---------|-----------|-----------|------------|----------|
| LoRA | 0.302 | 0.451 | — | 3.7s |
| ProLoRA | 0.281 | 0.443 | 0.961 | 3.7s |
| X-adapter | 0.271 | 0.403 | 0.884 | 16.1s |

## 5.7. Comparison with LoRA-X

This section compares the performance of transferred LoRA-X adapters (Farhadzadeh et al., 2025) against transferred standard LoRAs using our subspace and nullspace transfer (Equation 3). Table 12 provides a quantitative comparison across various datasets. Both LoRA and LoRA-X adapters are transferred from SDXL (source) to Stable Diffusion 1B (SSD-1B, target) and benchmarked against their respective counterparts trained from scratch on the target model. Transferred LoRA-X demonstrates slightly improved performance in certain cases. It is important to note the significant difference in rank between the two adapter types: LoRA uses rank 32, while LoRA-X uses rank 320. As discussed in Section 5.3.3, ProLoRA transfer is inherently lossy, and rank significantly impacts performance. Despite this, ProLoRA achieves comparable, underscoring the effectiveness of the subspace and nullspace projection method.

*Table 12.* Comparison of performance of transferred LoRA with rank 32 using ProLoRA with transferred LoRA-X with rank 320. The adapter is transferred from SDXL to SSD-1B.

| Datasets | Adapter | HPSv2 (↑) | LPIPS (↑) | DINOv2 (↑) | CSD-MMD (↓) |
|---|---|---|---|---|---|
| BlueFire | LoRA | 0.323 | 0.448 | 0.951 | **0.0207** |
| | ProLoRA | **0.318** | **0.413** | | |
| | LoRA-X | 0.316 | 0.428 | **0.969** | 0.0618 |
| | Tran LoRA-X | 0.300 | 0.392 | | |
| Paintings | LoRA | 0.328 | 0.436 | 0.946 | **0.0134** |
| | Pro LoRA | 0.318 | **0.433** | | |
| | LoRA-X | 0.319 | 0.409 | **0.961** | 0.0391 |
| | Tran LoRA-X | **0.320** | 0.355 | | |
| Origami | LoRA | 0.244 | 0.351 | **0.952** | 0.0245 |
| | ProLoRA | 0.256 | 0.343 | | |
| | LoRA-X | 0.244 | 0.412 | 0.941 | 0.0424 |
| | Tran LoRA-X | **0.269** | **0.388** | | |

## 5.8. Timing Comparison between different Transfer Methods

This section compares the time complexity of our proposed method, ProLoRA, with LoRA-X (Farhadzadeh et al., 2025). We benchmark performance transferring LoRAs between SDXL and SSD-1B. While LoRA-X boasts a faster transfer time of 92 seconds compared to ProLoRA's 271 seconds, due to ProLoRA's null space and full matrix SVD computations, LoRA-X requires training a specialized LoRA on the source model. This training takes significantly longer than standard LoRA training (0.2 iterations/second slower), resulting in an additional 400 seconds to reach convergence (over 2000 iterations). Therefore, despite faster transfer, the overall time overhead for LoRA-X adaptation significantly exceeds that of ProLoRA, which requires no source model training.

## 5.9. ProLoRA as Initialization

We experiment on finetuning the SSD-1B on the Dreambooth dataset with concept LoRA transferred from SDXL

*Table 13.* Wall clock time comparison when different adapters are trained on the source model SDXL and transferred to the target model SSD-1B. Measurements are done on 1 A100 GPU

| LoRA-X Train | LoRA Train | LoRA-X Transfer | ProLoRA Transfer |
|---|---|---|---|
| 2.3s/iter | 2.1s/iter | 92s | 271s |

vs random initalization. As shown in Table 14, using ProLoRA as initialization approach produces a large boost in performance. The performance of transfer at 250 iterations is also similar to that of random initialization at 1000 iterations.

*Table 14.* Performance sensitivity of ProLoRA to the chain of iterative transfer.

| Iteration | Initilization | CLIP-T | CLIP-I | DINOv2 |
|---|---|---|---|---|
| 250 | ProLoRA | 0.285 | 0.746 | 0.524 |
| | Random | 0.31 | 0.513 | 0.368 |
| 500 | ProLoRA | 0.291 | 0.749 | 0.549 |
| | Random | 0.287 | 0.602 | 0.431 |
| 750 | ProLoRA | 0.292 | 0.752 | 0.556 |
| | Random | 0.293 | 0.664 | 0.482 |
| 1000 | ProLoRA | 0.295 | 0.761 | 0.558 |
| | Random | 0.294 | 0.745 | 0.539 |

# 6. Conclusions

The increasing popularity of text-to-image diffusion models has spurred the adoption of PEFT techniques like LoRA, offering efficient fine-tuning with minimal parameter overhead. However, LoRA's inherent dependence on its base model necessitates retraining when new models emerge, often hampered by data availability constraints. ProLoRA, our proposed method, overcomes this limitation by enabling the direct transfer of LoRA adapters between diffusion models without retraining or requiring access to the original training data. By carefully mapping the LoRA's impact onto the subspace and nullspace of the target model's weights, ProLoRA preserves the adapter's functionality across different model architectures. Our experiments with text-to-image diffusion models demonstrate the efficacy of this approach, offering a practical solution for adapting LoRAs to evolving model landscapes while addressing data privacy and availability concerns. This work opens exciting avenues for future research, including extending ProLoRA to other PEFT methods and exploring its application in broader model adaptation scenarios.

## Impact Statement

This paper presents work whose goal is to advance the field of Machine Learning. There are many potential societal consequences of our work, none which we feel must be specifically highlighted here.

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

## A. Visualization Results using ProLoRA

Generated samples using ProLoRA-transferred style LoRAs are shown. Figures 9, 10, and 11 compare these transferred LoRAs (from SD-v1.5 to SD Eff-v1.0 and RV-v3.0) against LoRAs trained from scratch on BlueFire, Paintings, and Origami, respectively.

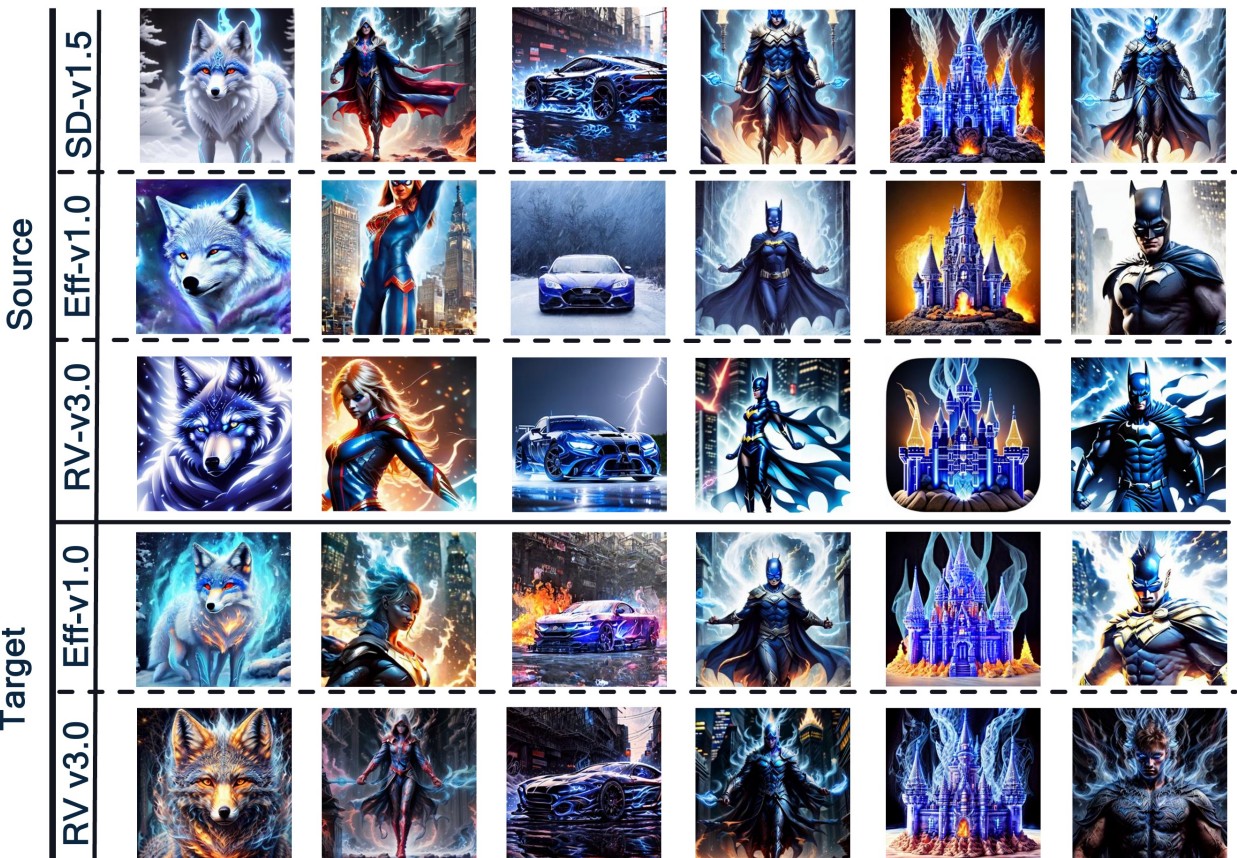

*Figure 9.* Generated samples using LoRA style adapter for BlueFire style on the SD-v1.5 as source model and ProLoRA training-free transfer to SD Eff-v1.0 and RV-v3.0. Results are also shown when SD Eff-v1.0 and RV-v3.0 are trained from scratch as the source model. Results are also shown when adapters on SD Eff-v1.0 and RV-v3.0 are trained from scratch as the source model. **Adapter**: "BlueFire", **Prompt**: 1) "wolf" 2) "girl in spiderman costume" 3) "car" 4) "woman in batman costume" 5) "castle in desert" 6) "man in batman costume".

## B. More Ablation Studies

### B.1. Impact of Null Space and Subspace

Figures 12a-c visualize the norm relationships between LoRAs trained on SDXL (source) and transferred to SSD-1B (target) using the Origami dataset. The strong correlation in overall norm (Figure 12a) suggests successful transfer, likely due to high subspace similarity. Further analysis, decomposing the norm into subspace and nullspace components (Figures 12b-c), reveals strong correlations in both, confirming the preservation of these components during transfer.

### B.2. No LoRA Baseline

We report the "No LoRA" baseline performance for the SD1.5-derived model family in Table 15. A comparison with Table 1 reveals that the CSD-MMD score for "No LoRA" is considerably poorer. This suggests that the baseline model without LoRA fails to capture the target style and, consequently, that our proposed CSD-MMD metric effectively detects style adaptation.

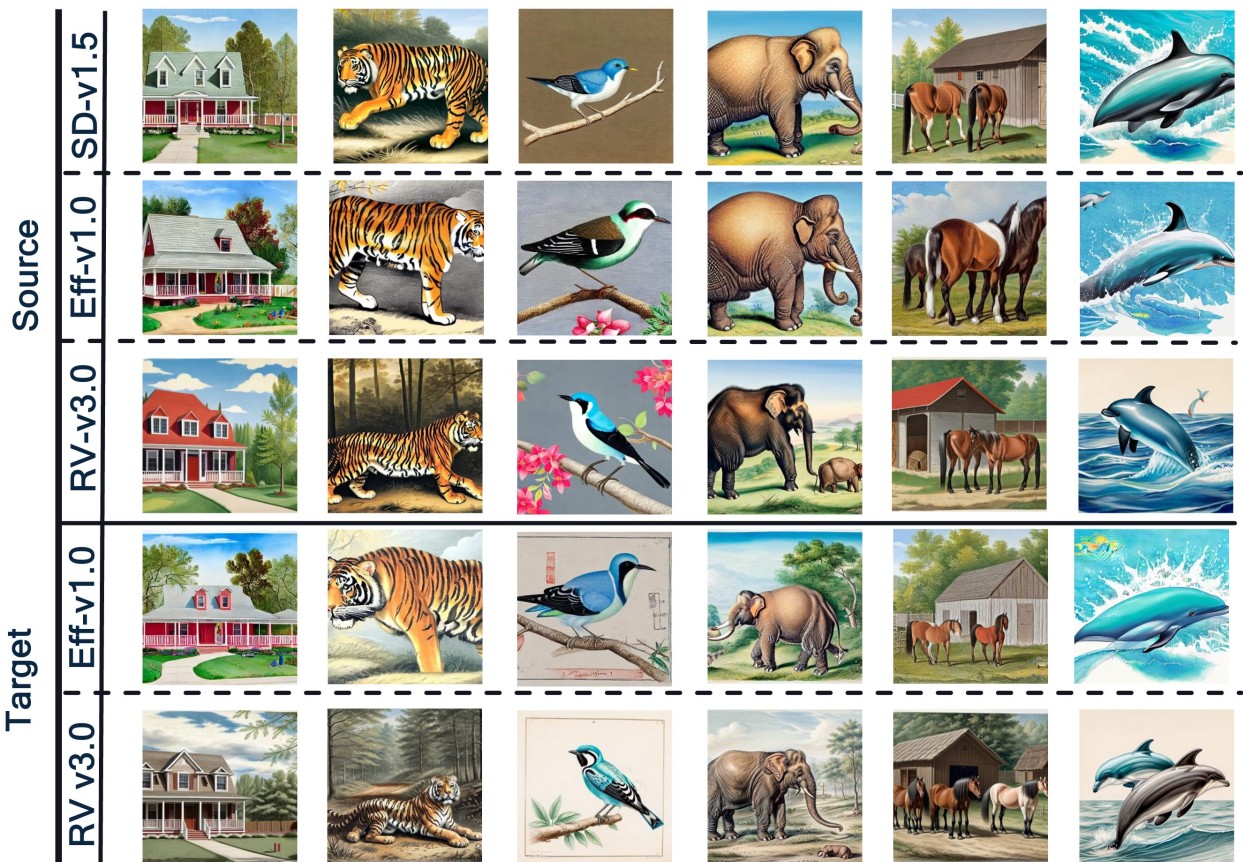

*Figure 10.* Generated samples using LoRA style adapter for Painting style on the SD-v1.5 as source model and ProLoRA training-free transfer to SD Eff-v1.0 and RV-v3.0. Results are also shown when SD Eff-v1.0 and RV-v3.0 are trained from scratch as the source model. **Adapter**: "Painting", **Prompt**: 1) "house on the prairie." 2) "tiger in the woods" 3) "bird on a tree branch" 4) "elephant in a grassland" 5) "horses eating grass, wooden hut" 6) "wild dolphins swimming".

*Table 15.* Comparison of text-to-image generation using LoRAs trained from scratch on target diffusion models versus training-free transfer using ProLoRA. LoRA rank is 32 for all cases.

| Datasets | Base Model | HPSv2 (↑) | LPIPS (↑) | CSD-MMD (↓) |
|---|---|---|---|---|
| BlueFire | SD Eff-v1.0 | 0.238 | 0.514 | 0.0071 |
| | RV-v3.0 | 0.279 | 0.498 | 0.0084 |
| Paintings | SD Eff-v1.0 | 0.284 | 0.518 | 0.0038 |
| | RV-v3.0 | 0.311 | 0.443 | 0.0049 |
| Origami | SD Eff-v1.0 | 0.239 | 0.501 | 0.0079 |
| | RV-v3.0 | 0.284 | 0.432 | 0.0065 |

## B.3. Qualitative Results

Figure 13 compares generated origami samples from SSD-1B using different methods: (1) style LoRA trained from scratch (baseline), (2) ProLoRA transferred from SDXL, (3) ProLoRA without nullspace projection, (4) copied LoRA with subspace similarity, and (5) copied LoRA without subspace similarity. Comparing rows 2 and 3 to the baseline (row 1) reveals that omitting nullspace projection affects 3D appearance. Rows 4 and 5 demonstrate that directly copying LoRAs produces distorted images compared to both the baseline and ProLoRA. Using subspace similarity (row 4) improves results, generating more diverse and origami-like images.

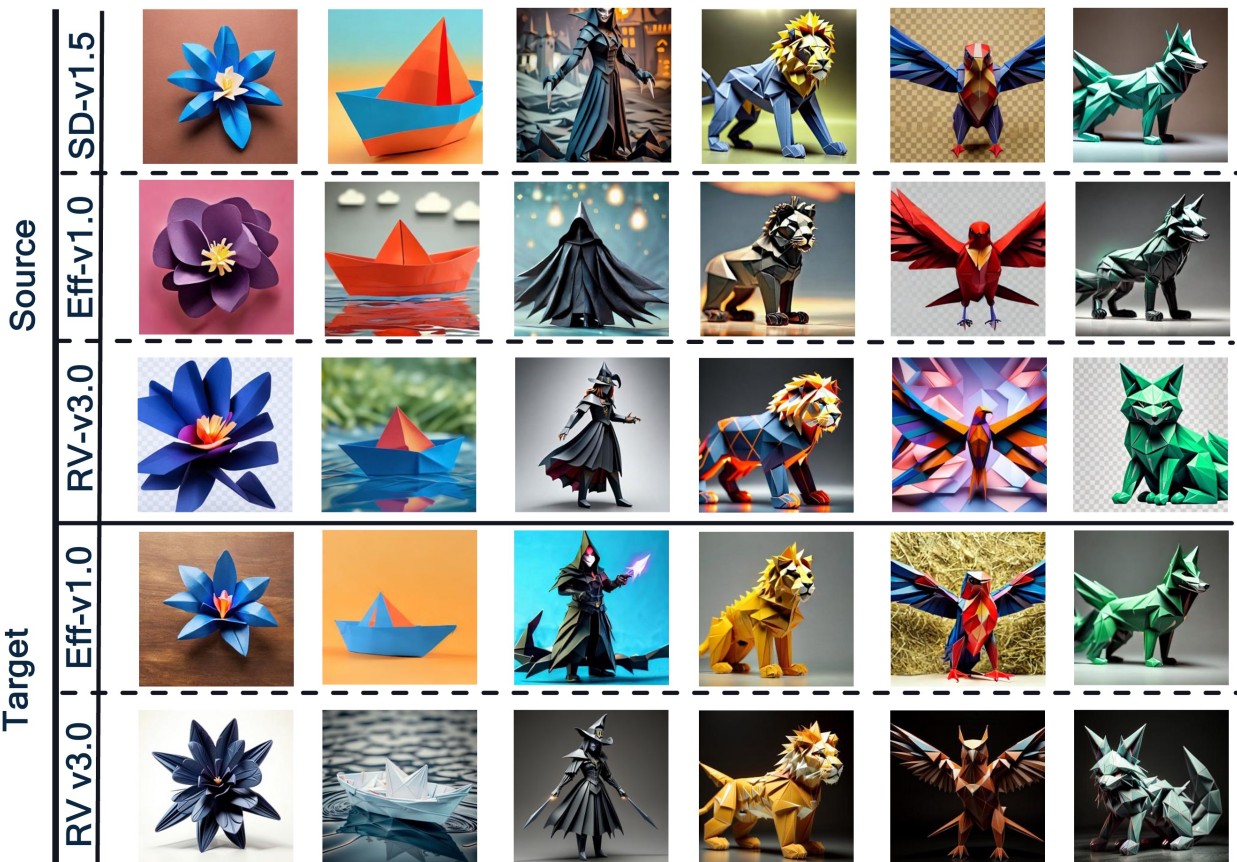

*Figure 11.* Generated samples using LoRA style adapter for Origami style on the SD-v1.5 as source model and ProLoRA training-free transfer to SD Eff-v1.0 and RV-v3.0. Results are also shown when SD Eff-v1.0 and RV-v3.0 are trained from scratch as the source model. Results are also shown when adapters on SD Eff-v1.0 and RV-v3.0 are trained from scratch as the source model. **Adapter**: "Origami", **Prompt**: 1) "flower" 2) "boat" 3) "medieval witch" 4) "lion" 5) "bird" 6) "fox".

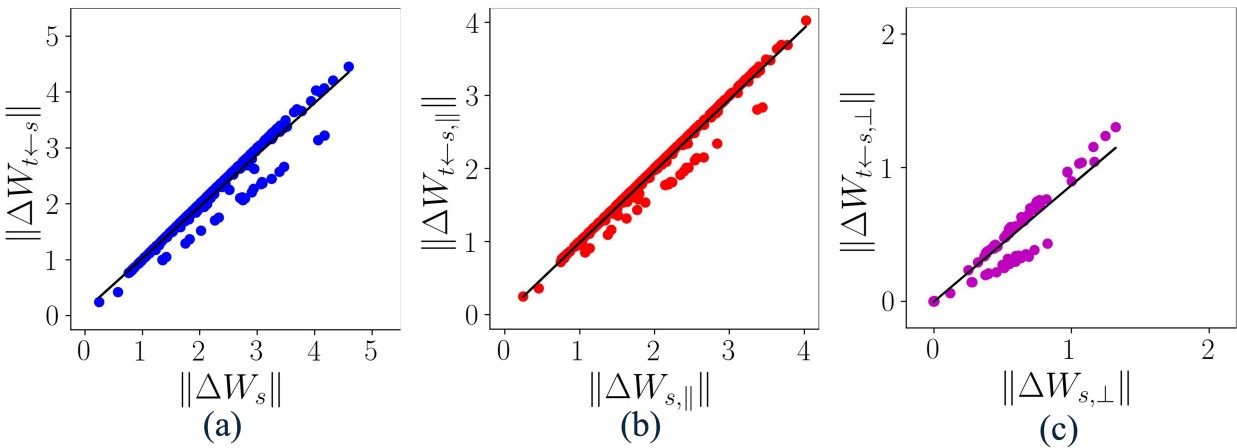

*Figure 12.* Correlation between SDXL (source) LoRA and transferred LoRAs (ProLoRA) to SSD-1B (target): (a) full LoRA norms, (b) subspace components, and (c) nullspace components.

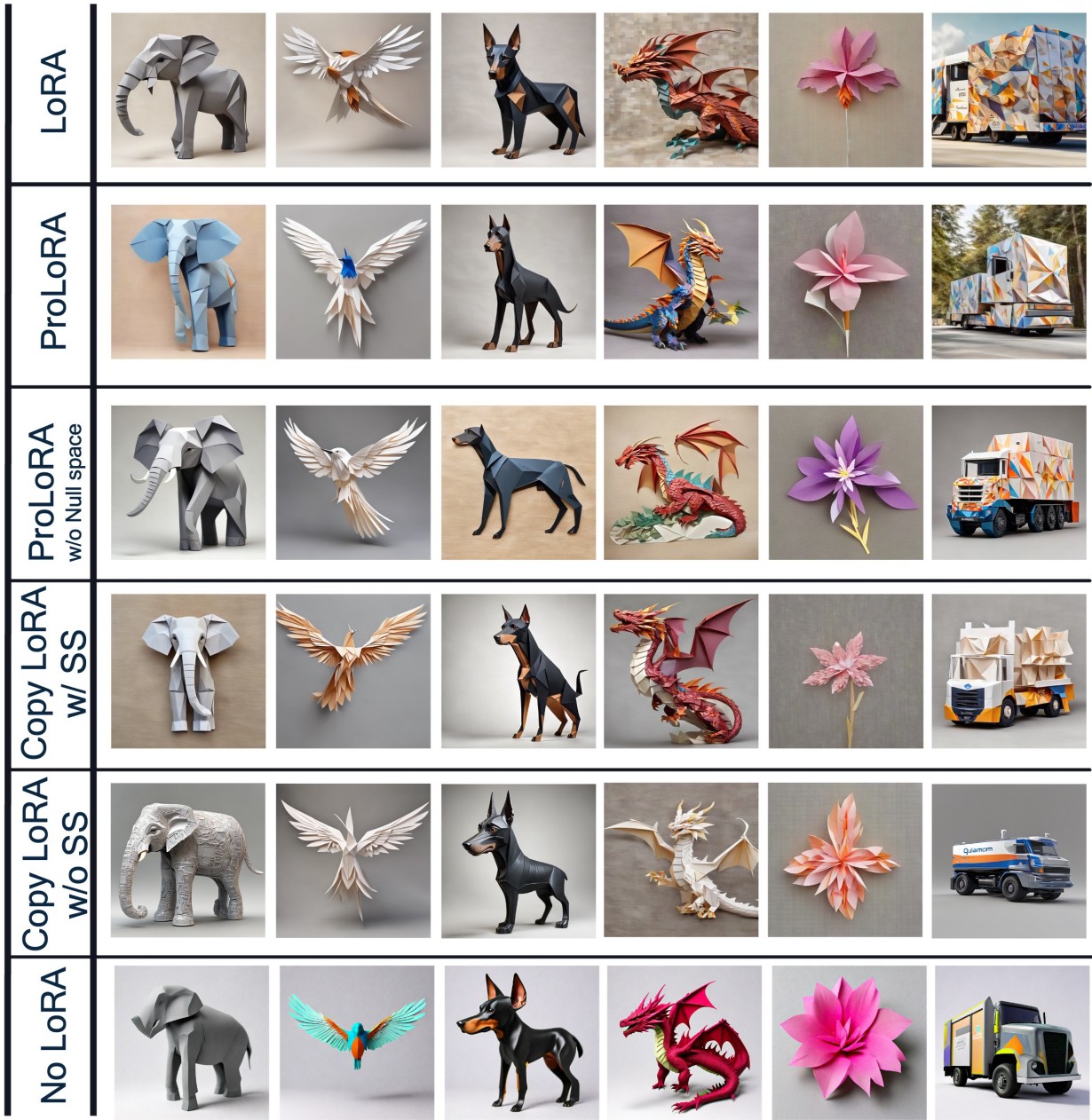

*Figure 13.* Effect of null space projection in ProLoRA by comparing samples genrated by SSD-1B using: (first row) style LoRA trained from scratch, (second row) style ProLoRA transferred from SDXL and (third row) ProLoRA without considering the null space projection. The forth row shows samples generated by SSD-1B using LoRA copied naively from SDXL. Adapter: "Origami", Prompt: 1) "elephant" 2) "bird with spread wings" 3) "doberman dog" 4) "dragon" 5) "flower" 6) "truck".

## C. Qualitative Results transferred DoRA

Figure 14 visualizes Origami dataset results, comparing generated samples using DoRA trained on SDXL and SSD-1B against DoRA transferred from SDXL to SSD-1B using ProLoRA.

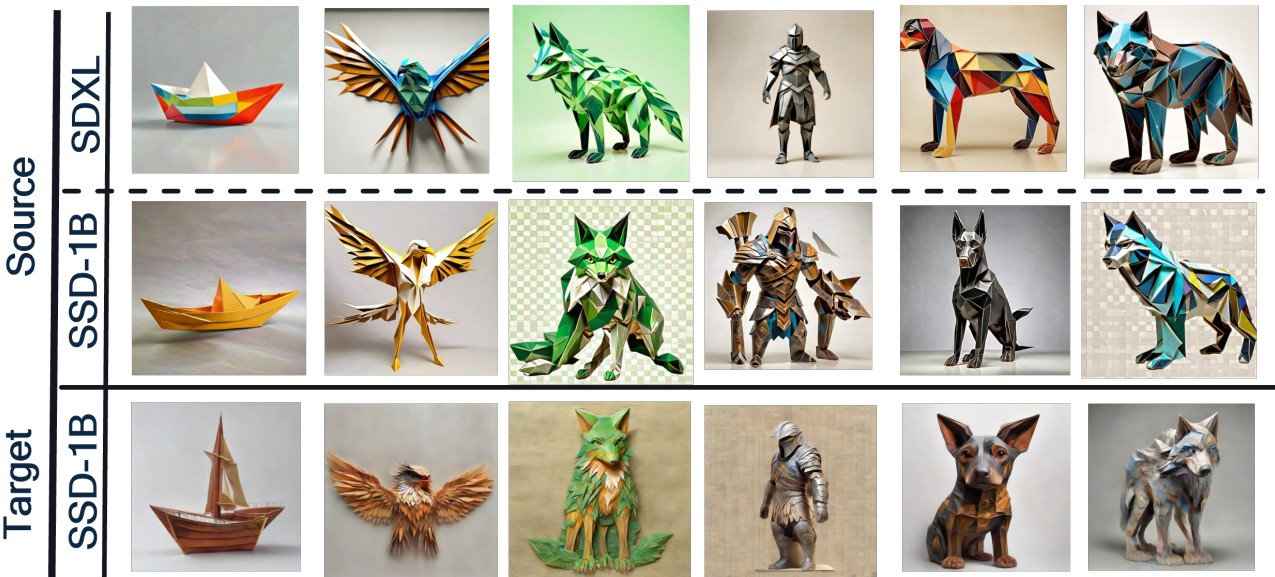

*Figure 14.* Generated samples using DoRA style adapter for origami style on the SDXL as source model and ProLoRA training-free transfer to SSD-1B. Results are also shown when DoRA is trained on SSD-1B from scratch as the source model. **Adapter**: "Origami", Prompt: 1) "boat" 2) "bird with spread wings" 3) "green fox" 4) "gladiator" 5) "doberman dog" 6) "wolf".

## D. Qualitative Results transferred FouRA

Figure 15 visualizes paintings dataset results, comparing generated samples using FouRAs trained on SD-v1.5 and RV-v3.0 against FouRAs transferred from SD-v1.5 to RV-v3.0 using ProLoRA.

## E. Experimental Setup for Text Generation

We implemented ProLoRA to fine-tune TinyLlama (Zhang et al., 2024) and successfully transferred adapters from TinyLlama 3T to TinyLlama 2.5T. We evaluated ProLoRA's transferability for different adapter types (LoRA and VeRA) on two standard text generation benchmarks from the original LoRA paper (Hu et al., 2022): text-to-text generation on the E2E NLG dataset (Novikova et al., 2017) (Table 16) and text summarization on the SamSum dataset (Gliwa et al., 2019) (Table 17). In both tasks, we observed only minor differences in BLEU and ROUGE scores between ProLoRA adaptations trained from scratch on the target model versus those transferred from the source model or directly copied. These results demonstrate ProLoRA's potential for efficient knowledge transfer across different language tasks and model variants.

*Table 16.* Evaluation of LoRA and VeRA trained from scratch on the base model TinyLlama 2.5T versus training-free transferred using ProLoRA from the source model TinyLlama 3T to the target model TinyLlama 2.5T in a text-generation task using the E2E-NLG dataset.

| Adapter | Method | ROUGE-1 (↑) | ROUGE-2 (↑) | ROUGE-L (↑) | ROUGE-LSum (↑) |
|---------|--------|-------------|-------------|-------------|----------------|
| LoRA | Trained | 0.7882 | 0.6341 | 0.7692 | 0.7634 |
|  | Transferred | 0.7881 | 0.6340 | 0.7684 | 0.7642 |
|  | Copied | 0.7634 | 0.6123 | 0.7482 | 0.7421 |
| VeRA | Trained | 0.7764 | 0.6224 | 0.7524 | 0.7532 |
|  | Transferred | 0.7782 | 0.6343 | 0.7620 | 0.7621 |
|  | Copied | 0.7544 | 0.6136 | 0.7481 | 0.7425 |

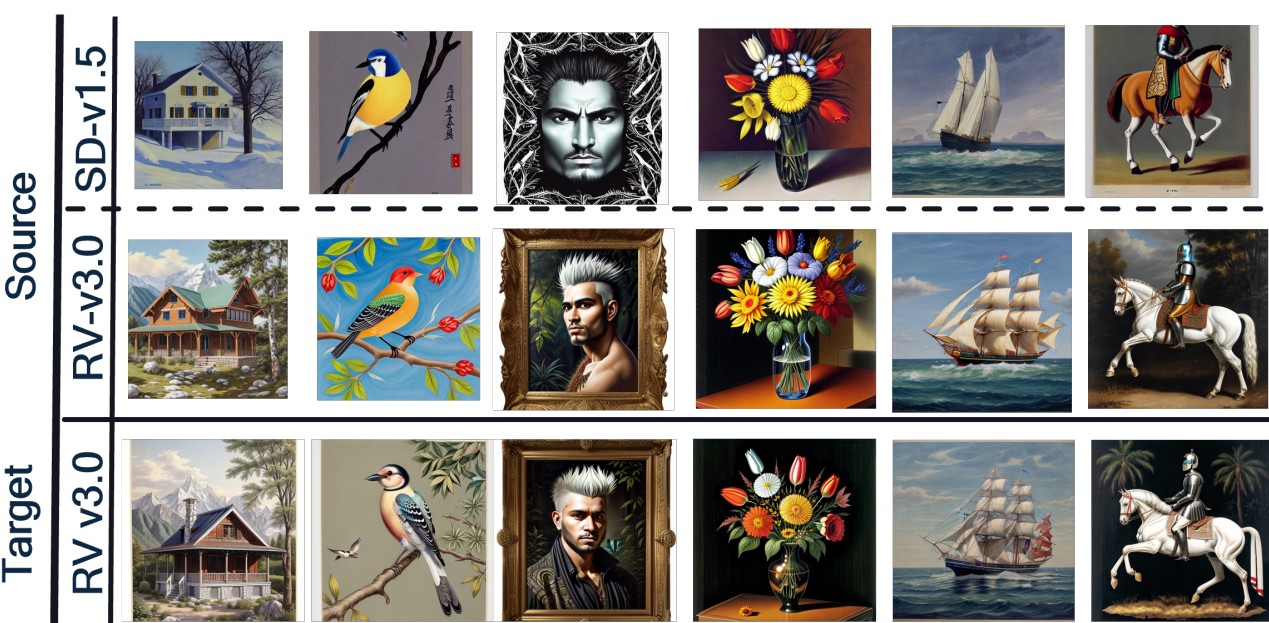

*Figure 15.* Generated samples using FouRA style adapter for paintings style on the SD1.5 as source model and training-free transfer to RV3.0. Results are also shown when FouRA is trained on RV3.0 from scratch as the source model. **Adapter:** "Painting", 1) "house on the Mountains." 2) "bird on a tree branch" 3) "man in a mythical forest, masterpiece, perfect face, intricate details, spiked hair" 4) "night flowers in vase, table " 5) "Ship sailing on sea" 6) "knight on a horse".

*Table 17.* Evaluation of LoRA and VeRA trained from scratch on the base model TinyLlama 2.5T versus training-free transferred using ProLoRA from the source model TinyLlama 3T to the target model TinyLlama 2.5T in a text-generation task using the SamSum dataset.

| Adapter | Method | ROUGE-1 (↑) | ROUGE-2 (↑) | ROUGE-L (↑) | ROUGE-LSum (↑) |
|---|---|---|---|---|---|
| LoRA | Trained | 0.3461 | 0.1596 | 0.2832 | 0.2862 |
| | Transferred | 0.3432 | 0.1546 | 0.2834 | 0.2852 |
| | Copied | 0.3213 | 0.1422 | 0.2623 | 0.2642 |
| VeRA | Trained | 0.3324 | 0.146 | 0.2722 | 0.2759 |
| | Transferred | 0.3312 | 0.1422 | 0.2712 | 0.2752 |
| | Copied | 0.3222 | 0.1402 | 0.26 | 0.2612 |

