# OpenReview forum: "Zero-Shot Adaptation of Parameter-Efficient Fine-Tuning in Diffusion Models"
_ICML.cc/2025/Conference — ICML 2025 poster_

### Official Review · Reviewer_qRQW · 2025-03-09

**Overall Recommendation:** 4

**Summary:**

The paper introduces **ProLoRA**, a parameter-efficient method for model adaptation and cross-domain knowledge transfer by decomposing pre-trained weight matrices via Singular Value Decomposition (SVD). ProLoRA splits a weight matrix W into a dominant subspace and a null-space. Key contributions include a low-rank subspace-zero-space decomposition framework, efficient adaptation through partial parameter updates, and empirical validation showing competitive performance in transfer learning tasks compared to LoRA and LoRA-X.

**Claims And Evidence:**

Most claims are supported by strong experiments justiﬁcation.

However, I still have some concerns about the empirical evidence of the semantics between the  subspace and null-space. In detail, the main contribution of this work is to decompose the weight into two orthogonal parts, thus facilitating the desired knowledge transfer. However, There lacks visualization results about what the specific meaning of this two parts and further include these explanation study can further justify the motivation.

**Essential References Not Discussed:**

The method have already discussed the latest baseline LoRA-X in detail.

**Experimental Designs Or Analyses:**

- The author experimented their methods on FouRA, it is recommended to apply more variants including VeRA, SVDiff and DoRA to verify the effectiveness  of the proposed method.


- The paper tests the LoRA transfer between source and target domain with limited pairs, which may weaken the generalization ability of the proposed method.

**Methods And Evaluation Criteria:**

- The detail of the decomposition is not clear. For instance, how the U matrix is decomposed into Us,∥ Us,⊥ needs more details. Moreover, why the first equal sign in  Eq.2  holds needs more explanation.

- The evaluation criteria mostly make sense for the problem or application at hand. It is suggested to include other common metrics in image generation, such as FID, CLIPscore.

- Since the methods include SVD within each layer, it is suggested to give the real runtime results instead of complexity analysis.

**Other Comments Or Suggestions:**

Typo in Line 119. "The right singular matrix U_s" -> "The Left singular matrix U_s"

**Other Strengths And Weaknesses:**

No further Strengths And Weaknesses.

**Questions For Authors:**

- How well does ProLoRA perform across other types of tasks or domains, like NLP or time-series analysis?

- Would additional fine-tuning be needed to adapt it effectively?

**Relation To Broader Scientific Literature:**

It is novel to project the LoRA weight into subspace and nullspace for transferring, which can inspire future works.

**Theoretical Claims:**

This paper does not include proofs.

---

> ### Author Rebuttal · Authors · 2025-04-01
>
> We thank the reviewer for valuable feedback and comments. Below, we provide detailed responses.
>
> [C1] However, I still  .... further justify the motivation.
>
> [R1] We agree that visualizing the subspace and nullspace's roles would strengthen our justification. We will include: (a) Semantic Ablation: Ablating each component and visualizing impact on style, content, etc. (b) Qualitative Examples: Showcasing distinct component contributions in transfer scenarios. These will concretely demonstrate each projection's semantic role, justifying ProLoRA's motivation.
>
> [C2] The detail of decomposition …. needs more explanation.
>
> [R2] We appreciate reviewer's request for more detail regarding decomposition process. The SVD yields left and right singular vectors of a matrix M. The right-singular vectors corresponding to vanishing singular values of ⁠ M⁠ span null space of ⁠M⁠, and left-singular vectors corresponding to the non-zero singular values of ⁠M span the range of ⁠M [1].
>
> [1] https://en.wikipedia.org/wiki/Singular_value_decomposition
>
> [C3] The evaluation criteria …. such as FID, CLIPscore.
>
> [R3] We appreciate suggestion to include FID and CLIPscore. We chose HPSv2 over CLIPscore because it's more sensitive to subtle improvements, especially in style/concept tasks. FID requires thousands of samples for robust estimate.
>
> [C4] Since the methods …. instead of complexity analysis.
>
> [R4] We understand reviewer's request for real runtime results given inclusion of SVD in each layer. We provide a computational complexity analysis in our response to reviewer eBAK [C7]. To summarize, while SVD is computationally intensive, ProLoRA performs SVD only once per model. The increased runtime is therefore limited to the initial decomposition, while subsequent adapter transfers benefit from the precomputed SVD. .
>
> [C5] The author experimented … FouRA … to verify the effectiveness of the proposed method.
>
> [R5] We appreciate the reviewer's suggestion to evaluate ProLoRA with more adapter variants. We have already conducted experiments with both FouRA (Table 8) and DoRA (Table 7) in the paper. SVDiff is similar to LoRA-X, but we acknowledge its value for completeness and add it in the final version. VeRA is not feasible on Diffusion model due to U-Net architecture (different KQV matrix size), but we're adding it to our LLM experiments and results are shown below. This will provide a more complete evaluation.
>
> Results are shown below for E2E-NLG Dataset.
>
> | Method | Adapter     | ROUGE-1 | ROUGE-2 | ROUGE-L | ROUGE-LSum |
> |--------|-------------|---------|---------|---------|------------|
> | VeRA   | Trained     | 0.7764  | 0.6224  | 0.7524  | 0.7532     |
> |        | Transferred | 0.7782  | 0.6343  | 0.762   | 0.7621     |
> |        | Copied      | 0.7544  | 0.6136  | 0.7481  | 0.7425     |
>
> The second results are for the SAM-SUM dataset
>
> | Method | Adapter     | ROUGE-1 | ROUGE-2 | ROUGE-L | ROUGE-LSum |
> |--------|-------------|---------|---------|---------|------------|
> | VeRA   | Trained     | 0.3324  | 0.146   | 0.2722  | 0.2759     |
> |        | Transferred | 0.3312  | 0.1422  | 0.2712  | 0.2752     |
> |        | Copied      | 0.3222  | 0.1402  | 0.26    | 0.2612     |
>
> For both datasets, it is seen that transferring VeRA v/s training VeRA from scratch for both the E2E-NLG and SAMSUM dataset produces very similar results suggesting that transferring VeRA using our proposed range space and null space projection is effective.
> Furthermore, copying is not effective as our proposed projection method.
>
> [C6] The paper tests LoRA transfer … of the proposed method.
>
> [R6] We acknowledge the reviewer's concern regarding the limited number of source-target pairs used in our evaluation, which could impact the perceived generalizability of ProLoRA. While our approach relies on a reasonable degree of subspace similarity between source and target models, and this similarity can be influenced even by models from different families, as shown in the paper many source-target pairs still meet this requirement.
>
> [C7] Typo in Line 119 …….
>
> [R7] Thanks, we'll correct that.
>
> [C8] How well does ProLoRA … or time-series analysis?
>
> [R8] We've expanded our evaluation to include tasks beyond image generation. Please refer in response to [C5] and reviewer eBAK's comment [C3].
>
> [C9] Would additional fine-tuning … effectively?
>
> [R9] To ensure we address the reviewer's intent, we would like to clarify whether the question is: "Can ProLoRA transfer be an effective initialization for subsequent fine-tuning?" We propose DreamBooth experiments comparing convergence and final performance (measuring [metrics]) with ProLoRA vs. random initialization. Please confirm if this aligns with your intent.

---

> > ### Comment · Reviewer_qRQW · 2025-04-02
> >
> > Thanks for providing this detailed rebuttal.
> >
> > After reading this response, most of my concerns have been addressed. It is suggested the authors to further revise the paper according to the discussion. I will upgrade my score.

---

> > > ### Author Response · Authors · 2025-04-09
> > >
> > > Dear Reviewer,
> > >
> > > Thanks for considering our rebuttal and improving the score. The paper will be revised according to the comments and responses.

---

### Official Review · Reviewer_xCjA · 2025-03-10

**Overall Recommendation:** 2

**Summary:**

This paper proposes ProLoRA, a zero-shot method for transferring pre-trained LoRA adapters between different text-to-image diffusion models without requiring retraining or access to original training data. The key motivation is that traditional LoRA adapters are tied to specific base models, making them difficult to reuse when models are updated/pruned/distilled, etc. ProLoRA consist of three stages of identifying module pairs, decomposing exist LoRA and transferring to the new model’s LoRA.

## update after rebuttal

I appreciate the authors’ rebuttal. However, after carefully reviewing their response, I still have concerns that remain unaddressed. Additionally, several important details, evaluations, and comparisons I highlighted in my initial review are critical and still absent from the manuscript. While this paper holds strong potential for publication, I believe it is not yet ready in its current form. Therefore, I will maintain my original score.

**Claims And Evidence:**

Partially. While the paper clearly defines its task, the experimental validation appears incomplete due to the absence of critical baselines and a lack of clarity in the evaluation metrics.

Baseline Comparisons:

The majority of comparisons in the paper, including both results and ablation studies, are limited to the LoRA baseline (e.g., Tables 1, 3, 4, 6, and 7). However, additional baselines are essential to properly assess the contribution of the proposed method.
For instance, the evaluation should include the original model without any adapter to determine whether ProLoRA provides a meaningful improvement over the base model. Additionally, a naive transfer of the source LoRA (just a copy) should be included to verify whether ProLoRA truly enhances transferability or if a simple weight copy already performs comparably.
Consider Table 1, where ProLoRA is only 0.04 away from the LoRA baseline. Is this a substantial improvement, or is it within the range of natural variation? If the original model (without any adapter) produces similar results, then ProLoRA’s impact may be negligible. Likewise, if directly transferring the source LoRA (despite being trained on a different model) yields comparable or better performance, it would call into question the necessity of the proposed method. These open questions make it difficult to fully assess the contribution of ProLoRA, and addressing them with appropriate baselines would significantly strengthen the paper’s claims.

In contrast, Table 2 does include adapter-free baseline models (last two rows), but surprisingly, these baselines achieve higher HPSv2 and LPIPS scores than the fine-tuned LoRA and ProLoRA. This is unexpected - why does the adapted model underperform compared to an adapter-free model? This discrepancy should be addressed, as it calls into question whether adaptation was achieved with LoRA/ProLoRA.

Metric Unclarities:

The paper provides a brief explanation of CSD-MMD (lines R189-R200), but fails to specify the reference image sets used for calculation. Was CSD-MMD computed against images from the original model, the LoRA-adapted model, or the training dataset? I assume it was the dataset, but the missing values from Tables 4-5 create ambiguity.

Table 3 introduces CLIP-T/I metrics, what are these metrics actually? Also, these are not used consistently across the paper. Why are these metrics included only for this experiment? Shouldn’t all tasks and ablations be evaluated using a uniform set of metrics? This inconsistency makes it difficult to compare results across different sections of the paper.
The paper would benefit from a clearer and more comprehensive set of baseline comparisons, along with a standardized set of evaluation metrics across experiments. Addressing these concerns would significantly improve the credibility of the claimed contributions.

**Essential References Not Discussed:**

The work of Wang et al., 2024 is cited incorrectly in line 070—the reference points to an arXiv version, while a published NeurIPS 2024 version exists. The authors should update the citation to reflect the peer-reviewed conference paper instead of the preprint. I recommend verifying all citations in the paper.

**Experimental Designs Or Analyses:**

Yes, please see above comments

**Methods And Evaluation Criteria:**

For diffusion models, yes, the proposed method and benchmarks seem appropriate. However, the generality of the method raises important questions.
The approach appears quite broad and potentially applicable beyond diffusion models, much like LoRA itself. This leads to a natural curiosity: how does ProLoRA perform on other tasks or backbones, such as:
- Large Language Models (LLMs) for general language understanding,
- Vision Transformers (ViTs) for image classification,
- Image-to-image generation models, etc.

Have the authors attempted applying their method to any of these domains? Even if the results were unsuccessful, discussing why ProLoRA is effective specifically for diffusion models (and not for other tasks/backbones) would be insightful. If ProLoRA has not been tested on other backbones, including such experiments - even as preliminary results would greatly enhance the paper’s impact and

**Other Comments Or Suggestions:**

- Typo line 119: “right” —> “left”
- Fig 1 is hard to see, I need a serious zoom-in to see any origami evidence. Better to choose another example or modify this one.
- Table 2 need to be re-arragned, it is hard to read it. There is only one cross-transfer experiment


An interesting potential extension of ProLoRA is its use as an initialization method for fine-tuning LoRA on a new model. If accuracy is a priority and re-training LoRA is inevitable, it would be valuable to explore whether initializing LoRA with ProLoRA (instead of default initialization) improves convergence speed and final performance.

**Other Strengths And Weaknesses:**

Building on my previous comments, the proposed method appears highly general, with significant potential impact beyond diffusion models. Its deterministic, simple, and training-free nature makes it particularly appealing. However, the paper misses an opportunity to evaluate ProLoRA on other tasks and model backbones, which could provide valuable insights for future research. Expanding the evaluation scope would not only strengthen the contribution but also help position ProLoRA as a more broadly applicable method in the field of adapter transfer and parameter-efficient fine-tuning.

**Questions For Authors:**

- What is CLIP-I and CLIP-T metrics?
- Why LoRA is the least performing in the HPSv2 metric in Tab 5?

**Relation To Broader Scientific Literature:**

The proposed method has the potential to impact a broad range of architectures, much like LoRA, given its focus on efficient adaptation. If validated beyond diffusion models, it could contribute to knowledge transfer across widely used LoRA adapters in various domains.

**Theoretical Claims:**

N/A

---

> ### Author Rebuttal · Authors · 2025-04-01
>
> We thank the reviewer for valuable feedback and comments.
>
> [C1] The majority of comparisons … strengthen paper’s claims.
>
> [R1] "No LoRA" can degrade performance (e.g., produce blurry outputs in Table 4). We are currently running experiments for Tables 1. We will add results of Tables 6 & 7 in the final version. For Table 3 we have added results below and we observe that “No LoRA” produces poor performance.
>
> Following are results on concept customization
>
> | Method    | CLIP-T | CLIP-I | DINOv2 |
> |-----------|--------|--------|--------|
> | No LoRA   | 0.251  | 0.521  | 0.352  |
> | LoRA      | 0.294  | 0.745  | 0.539  |
> | Copy LoRA | 0.300  | 0.719  | 0.475  |
> | ProLoRA   | 0.287  | 0.737  | 0.501  |
>
> [C2] In contrast, …  whether adaptation was achieved with LoRA/ProLoRA.
>
> [R2] There might be cases where HPSV2 and LPIPS score is higher for non-LoRA baseline. HPSV2 is prompt fidelity while LPIPS is diversity. Sometimes after fine-tuning on a particular style it produces similar looking images and hence LPIPS might be lower. Also, prompt fidelity might be reduced due to presence of certain style in the generated image. However, DINOV2 is still higher for cases where adapter is fine-tuned.
>
> [C3] The paper … Tables 4-5 create ambiguity.
>
> [R3] We apologize for the lack of clarity regarding CSD-MMD calculation.
> - General Use: We use CSD-MMD to compare generated samples from ProLoRA-transferred model against the generated samples from LoRA-trained model. This helps quantify the similarity of output distributions.
> - Table 4 (LCM-LoRA): CSD-MMD is not applicable to Table 4 because this table focuses on LCM-LoRA adaptation for accelerated sampling, not style transfer. CSD-MMD is primarily used to measure style similarity.
> - Table 5: In Table 5, CSD-MMD is computed against generated samples from LoRA-trained model (shown in the first row of the table).
> We will clarify these details in the final version of the paper.
>
> [C4] Table 3 introduces CLIP-T/I metrics, … significantly improve the credibility of claimed contributions.
>
> [R4] The reason for inconsistency is that our paper considers a range of tasks (style adaptation, concept customization, transfer of acceleration LoRAs), and each of these tasks has established evaluation practices in the literature. CLIP-I [1] and CLIP-T [1], for instance, are standard metrics in concept customization, measuring image and text fidelity, respectively. We will clarify all these details in the paper.
>
> [1] Radford, Alec, et al. "Learning transferable visual models from natural language supervision." ICML 2021.
>
> [C5] For diffusion models, .... enhance the paper’s impact and
>
> [R5]  As detailed in our response to reviewer eBAK's comment [C3], we have conducted preliminary experiments applying ProLoRA to the TinyLlama language model on the SamSum and E2E-NLG datasets
>
> [C6] The work of Wang et al., 2024 … all citations in the paper.
>
> [R6] Thank you for pointing out incorrect citation. We will update the reference to Wang et al., 2024 to the published NeurIPS 2024 version.
>
> [C7] Building on my previous comments, …. and parameter-efficient fine-tuning.
>
> [R7] We've expanded our evaluation to include language tasks beyond image generation. As detailed in response to reviewer qRQW's comment [C5] and reviewer eBAK's comment [C3]
>
> [C8] Typo line 119: “right” —> “left”
>
> [R8] Thank you for catching the typo. We will correct that.
>
> [C9] Fig 1 is hard to see, I need a serious zoom-in to see any origami evidence. Better to choose another example or modify this one.
>
> [R9] We appreciate the feedback … visual differences in the revised version.
>
> [C10] Table 2 need to be re-arranged, it is hard to read it. There is only one cross-transfer experiment
>
> [R10] We appreciate the feedback regarding the readability of Table 2. We re-arrange the table to improve its clarity and organization.
>
> [C11] An interesting potential extension of ProLoRA … improves convergence speed and final performance.
>
> [R11] This is a great suggestion! We'll conduct DreamBooth experiments comparing ProLoRA vs. default initialization (measuring [metrics]) and share results in the next round. Does this experiment align with your suggestion?
>
>
> [C12] What is CLIP-I and CLIP-T metrics?
>
> [R12] CLIP-I and CLIP-T are metrics based on the CLIP model, commonly used to evaluate concept customization. CLIP-I assesses image fidelity, while CLIP-T assesses text fidelity. They are standard metrics in this area.
>
> [C13] Why LoRA ...in Tab 5?
>
> [R13] HPSv2 measures prompt fidelity (image-text alignment). LoRA fine-tuning specializes the model towards a specific concept/style. This specialization leads LoRA to prioritize the learned concept over precise adherence to the full prompt text, resulting in a lower HPSv2 score. Different metrics emphasize different qualities – HPSv2 on prompt following, while others like CSD-MMD might focus more on aspects like style consistency where LoRA could perform better.
>
> **Hope responses suffice enough to raise score**

---

### Official Review · Reviewer_eBAk · 2025-03-15

**Overall Recommendation:** 4

**Summary:**

This paper introduces ProLoRA, a method for zero-shot transfer of LoRAs between source and target diffusion models. It features a projection technique that transfers both subspace and nullspace components of source LoRAs to target models while preserving generation performance. The method works by identifying similar modules between models, decomposing the LoRA and projecting the LoRA components into the target model's weight space. Evaluations across datasets and models show comparable performance to training from scratch.

## update after rebuttal
The authors addressed my concerns satisfactorily, therefore I will maintain my conditional Accept as final.

**Claims And Evidence:**

- The primary claim that the proposed work enables training-free transfer of LoRA adapters between models is well-supported by both qualitative and quantitative results.
- Both subspace and nullspace projections are necessary is demonstrated through ablations and visualizations, showing performance degradation when either component is removed.
- Appropriate metrics are convincing for the claim that similar performance is attained against LoRAs trained from scratch and other works.
- The generalization claim about different PEFT methods (DoRA, FouRA) is supported. However, fewer experiments are presented compared to the main LoRA evaluations.

**Essential References Not Discussed:**

All essential related works are discussed, to my knowledge.

**Experimental Designs Or Analyses:**

- The experiments on style transfer datasets appear sound, with appropriate comparisons and metrics. The experiments on concepts using DreamBooth data are convincing too. Finally, LCM-LoRA experiments about acceleration show the method versatility.
- The comparisons with competing methods (X-adapter, LoRA-X) are fair and highlight ProLoRA's advantages, although more extensive benchmarking would strengthen the claims.

**Methods And Evaluation Criteria:**

- The methodological approach is sound, with clear mathematical formulations of the projection operations for both subspace and nullspace components.
- The evaluation protocol comparing against LoRAs trained from scratch on target models provides a strong baseline for assessing transfer quality.
- The chosen metrics for evaluation makes sense for measuring image generation quality, diversity, and style transfer fidelity.
- The ablation studies are extensive and well-designed to understand the contributions of the proposed method.

**Other Comments Or Suggestions:**

- Consider adding more quantitative analysis of when module similarity matters most for successful transfer.
- The naming of metrics in tables could be more consistent throughout the paper.
- A table summarizing the computational requirements for different model transfers would help readers understand practical implications.

**Other Strengths And Weaknesses:**

### Strengths

- The method addresses a practical problem of reusing existing adapters. Given current artistic hubs of these, the proposed method can be potentially very useful.
- The nullspace component analysis reveals an often-overlooked aspect of LoRA adapters' functionality.
- The experiments across different adapter types demonstrates broad applicability.

### Weaknesses

- The method still requires computing SVD on both source and target models, which can be computationally expensive for very large models.
- The performance varies across different datasets and model pairs, this suggests some limitations in generalizability.
- Following previous point, the theoretical analysis could be deepened to provide better insights into when and why the method might fail.
- The evaluation is limited to text-to-image diffusion models, therefore testing on other modalities would strengthen the paper claims.
- Only convolution-based architectures were explored. Current state-of-the-art models are mainly transformer-based due to scalability.

**Questions For Authors:**

- How does the performance of ProLoRA degrade when the architectures of source and target models differ more substantially? For example, would the method work between completely different architectures like a SD1.5 to SDXL or SDXL to SD3?
- Could you provide more details on how the threshold of 0.8 for subspace similarity was selected? How sensitive is the method to this hyperparameter?
- How would the proposed method perform if applied iteratively across a chain of models $A \rightarrow B  \rightarrow C$ compared to direct transfer $A \rightarrow C$ ?

**Relation To Broader Scientific Literature:**

The paper contributions are related to transfer methods, including knowledge distillation approaches. It is close to PEFT literature and LoRA works. LoRA-X is particularly relevant as it shares similar goals but with key methodological differences. The work connects to broader subspace analysis techniques in neural networks, though this connection could be more explicitly developed.

**Theoretical Claims:**

I informally read through formulations in Sections 4.1-4.3, particularly equations (1)-(3), and look correct to me.
The computational complexity analysis of the initial SVD computation is correct.

---

> ### Author Rebuttal · Authors · 2025-04-01
>
> We thank the reviewer for valuable feedback and comments. Below, we provide detailed responses.
>
> [C1] The method still requires computing SVD … for very large models.
>
> [R1] We address this point in our response to Reviewer DiKB's comment [C5].
>
> [C2] The performance varies …. limitations in generalizability. Following previous point, … why the method might fail.
>
> [R2] We attribute performance variations to mismatches in: (1) Subspace Similarity (attention differences) and (2) Architectural Divergence. To address this, we will: (a) Incorporate a transferability metric (like LoRA-X Fig. 4) to predict transfer success. (b) Analyze failure cases to identify and mitigate limitations. (c) Refine our theoretical analysis to account for subspace similarity/architectural divergence.
>
> [C3] The evaluation is limited ... strengthen the paper claims.
>
> [R3] We thank the reviewer for highlighting the limited evaluation scope. To address this, we've included initial results of applying ProLoRA to TinyLLAMA model on SamSum and E2E-NLG datasets, benchmarks. These experiments demonstrate ProLoRA's applicability beyond image generation. Copying does not produce good performance.
>
> The first results are E2E-NLG
>
> | Method  | Adapter     | ROUGE-1 | ROUGE-2 | ROUGE-L | ROUGE-LSum |
> |---------|-------------|---------|---------|---------|------------|
> | ProLoRA | Trained     | 0.7882  | 0.6341  | 0.7692  | 0.7634     |
> |         | Transferred | 0.7881  | 0.6340  | 0.7684  | 0.7642     |
> |         | Copied      | 0.7634  | 0.6123  | 0.7482  | 0.7421     |
>
> The second results are for SamSum dataset
>
> | Method  | Adapter     | ROUGE-1 | ROUGE-2 | ROUGE-L | ROUGE-LSum |
> |---------|-------------|---------|---------|---------|------------|
> | ProLoRA | Trained     | 0.3461  | 0.1596  | 0.2832  | 0.2862     |
> |         | Transferred | 0.3432  | 0.1546  | 0.2834  | 0.2852     |
> |         | Copied      | 0.3213  | 0.1422  | 0.2623  | 0.2642     |
>
>
> [C4] Only convolution-based architectures ... due to scalability.
>
> [R4] Our text-to-image models incorporate transformer blocks for attending to text embeddings. Additionally, the Table above includes results for language models. These show ProLoRA's applicability to pure transformer-based architectures.
>
> [C5] Consider adding more ... for successful transfer.
>
> [R5] To address reviewer's request for more quantitative analysis on the impact of module similarity for successful transfer, we wanted to clarify whether analyzing effect of transferring not only correlated modules (as in Table 5) but also non-correlated modules would suffice. Upon confirmation, we proceed to perform those experiments.
>
> [C6] The naming of .... throughout the paper.
>
> [R6] We will improve our convention in the paper.
>
> [C7] A table... understand practical implications.
>
> [R7] As suggested, please refer Table 11 of Appendix. Additionally , the transfer and inference process of X-adapter is same of 17.1s.  Finetuning time might be estimated to be pretty long due to large scale training.
>
> [C8] How does performance of ProLoRA .... a SD1.5 to SDXL or SDXL to SD3?
>
> [R8] While ProLoRA is designed for training-free transfer within same architectural family, its effectiveness may decrease with greater divergence. Exploring these limits like transferring across different architectures is an interesting future direction.
>
> [C9] Could you provide ... threshold of 0.8 for subspace similarity ...? How sensitive ... hyperparameter?
>
> [R9] The initial threshold of 0.8 for subspace similarity was chosen based on empirical analysis. To assess sensitivity of ProLoRA to this hyperparameter, we conducted experiments with thresholds of 0.9 and 1.0 when transferring LoRA from SDv1.5 to Eff v1.0.  These initial results suggest that ProLoRA is relatively robust to variations in threshold. We plan to include more comprehensive results in final version.
>
> | Method           | Dataset | Threshold | CSD-MMD |
> |------------------|-------|-------|---------|
> | ProLoRA          | Blue fire | 0.8 | 0.0025  |
> | ProLoRA          | Blue fire | 0.9 | 0.0031  |
> | ProLoRA          | Blue fire | 1.0 | 0.0082  |
>
> [C10] How would proposed method .... compared to direct transfer?
>
> [R10] To assess iterative transfer, we compared chained transfers (SD1.5 -> RV3 -> EffNet v1.0) against direct transfer (SD1.5 -> EffNet v1.0). Results indicate that iterative transfer degrade performance (Origami dataset), potentially due to error accumulation. We will explore this further with more models/datasets in final version.
>
> | Dataset  | CSD-MMD (SD-1.5 -> Eff v1.0 ) | CSD-MMD (SD-1.5 -> RV-3 -> Eff v1.0 ) |
> |----------|-------------------------------|---------------------------------------|
> | Painting | 0.0026                        | 0.0027                                |
> | Origami  | 0.0025                        | 0.0045                                |
> | Bluefire | 0.0025                        | 0.0025                                |

---

> > ### Comment · Reviewer_eBAk · 2025-04-07
> >
> > I thank the authors for their work and for addressing my comments. I am satisfied with the response and will keep my score.

---

> > > ### Author Response · Authors · 2025-04-09
> > >
> > > Dear Reviewer,
> > >
> > > Thanks for acknowledging our rebuttal and keeping the score.

---

### Official Review · Reviewer_DiKB · 2025-03-16

**Overall Recommendation:** 3

**Summary:**

The paper proposes ProLoRA, which can transfer the pre-trained LoRA to another target model without training. This addresses a key constraint in existing methods where LoRA adapters are trained to specific models, requiring complete retraining to a new model. ProLoRA projects source to target weight space by utilizing subspace and null space similarities and selectively targeting aligned layers. The proposed method is evaluated on diverse image generation tasks with diffusion models.

## update after rebuttal
Thank you to the authors for their response. As most of my concerns have been sufficiently addressed, I am increasing my score to a weak accept.

**Claims And Evidence:**

The central claim of this paper is that ProLoRA enables effective zero-shot adaptation of parameter-efficient fine-tuning across different text-to-image diffusion models by critically incorporating both subspace and null space projections. Unlike previous approaches such as LoRA-X, which only considers subspace projection, ProLoRA's key innovation is its comprehensive projection methodology that preserves the full expressiveness of the source LoRA adapter. At the same time, the paper emphasizes that the decomposition and projection process used in ProLoRA can be executed significantly faster than training a new LoRA adapter from scratch on the target model. These are supported by the experimental section.

**Essential References Not Discussed:**

References are well discussed.

**Experimental Designs Or Analyses:**

The authors evaluate their methodology using various diffusion models including SDXL, Stable Diffusion v1.5, and SSD-1B, as well as different adapter types such as style, concept, and LCM-LoRA. Their assessment combines both quantitative metrics and qualitative visual examples to demonstrate the effectiveness of their approach.

**Methods And Evaluation Criteria:**

The method is also evaluated on diverse generation tasks HPSv2, LPIPS, and CSD-MMD scores. Additionally, DINOv2 score is also used measure similarity between source and target generation. Wall-clock time is also used to compare time effiiciency.

**Other Comments Or Suggestions:**

It would be great to provide more analysis on the role of each subspace or null space projection. While the paper effectively demonstrates their empirical importance, a deeper analytical examination of their distinct contributions would strengthen the theoretical foundation. Additional insights into how each projection component preserves specific visual features or stylistic elements would enhance understanding of the transfer mechanism.

**Other Strengths And Weaknesses:**

Strength:
- Different from LoRA-X, the method does not require additional training of LoRAs on the source model.
- The method additionally leverages nullspace transfer.
- The proposed method is evaluated on diverse backbones and transfer scenarios.


Weakness:
- The technical contribution is very marginal compared to LoRA-X.
- The performance appears to be underwhelming.
- The baseline for the proposed method is not clear. It seems like LoRA-X is the closest work and should be the baseline for all experiments. However, only a part of the experiments explores this comparison.
- The paper claims the disadvantage of LoRA-X is that it requires pre-training. However, it is the same for ProLoRA that requires LoRA trained on a source model.
- The full SVD computation has a high computational cost, with minimal benefits over the baselines.

**Questions For Authors:**

Please refer to the weakness section.

**Relation To Broader Scientific Literature:**

Its contribution could be extended to other areas other than image generation.

**Theoretical Claims:**

The authors provide theoretical insights for their approach, using decomposition to decompose weights into subspace and null space components. The equations for projecting the source LoRA onto both spaces of the target model (Equations 2 and 3) are well-explained. While the paper demonstrates empirically that the null space projection is crucial through ablation studies, the theoretical justification for why standard LoRAs affect the null space is somewhat underdeveloped.

---

> ### Author Rebuttal · Authors · 2025-04-01
>
> We thank the reviewer for valuable feedback and comments. Below, we provide detailed responses.
>
> [C0] Its contribution could be extended to other areas other than image generation.
>
> [R0] Please refer [C3] of Reviewer eBaK
>
> [C1] The authors provide theoretical insights ...  why standard LoRAs affect the null space is somewhat underdeveloped.
>
> [R1] While we appreciate the feedback regarding the theoretical justification for standard LoRAs affecting the null space, we understand the concern as follows: Non-square weight matrices have inherent nullspaces. Unlike LoRA-X and SVDiff, standard LoRA finetuning doesn't constrain updates to the model's weight subspace; it allows modifications in both the weight and nullspaces, potentially causing unintended effects. This is because standard LoRA lacks the constraint of only optimizing singular values. Therefore, we propose projecting LoRA updates into both range and nullspaces during transfer to mitigate these effects.
>
> [C2] The technical contribution is very marginal compared to LoRA-X.
>
> [R2] We acknowledge that ProLoRA builds upon existing research, including LoRA-X. However, ProLoRA's approach offers distinct advantages that make it more versatile and broadly applicable. LoRA-X, by modifying only the singular values of the pre-trained model weights, effectively restricts the adapter to the weight subspace and ignores the nullspace. This constraint necessitates higher adapter ranks (e.g., 320 for LoRA-X versus 32 for standard LoRA), leading to increased inference computation. Furthermore, LoRA-X adapters are only transferable from other LoRA-X-trained models. ProLoRA, conversely, allows transfer from diverse, pre-existing adapters. Since ProLoRA isn't constrained to the pre-trained weight subspace during training, its updates are decomposed into range and null space components and then projected onto the target modules, enabling greater flexibility.
>
>
> [C3] The baseline for the proposed method ... only a part of the experiments explores this comparison.
>
> [R3] We evaluated the performance of a target model using two approaches: first, training a LoRA adapter from scratch with access to the training dataset, and second, transferring a LoRA adapter from a source model to the target model using ProLoRA. The performance of the LoRA adapter trained from scratch serves as an upper-bound baseline, representing the ideal performance we aim to achieve through transfer learning. We had included LoRA-X as a baseline for the style dataset in Table 10. LoRA-X comparison on LCM-LoRA models do not make sense as we use pretrained acceleration modules. Applying LoRA-X on the Dreambooth dataset for concept customization shows very poor performance and does not converge. This is likely due to the fact the LoRA-X only updates singular values which is not representative enough to add new concepts to the model.
>
> [C4] The paper claims the disadvantage of  LoRA-X .... ProLoRA that requires LoRA trained on a source model.
>
> [R4] We acknowledge the reviewer's point that ProLoRA, like LoRA-X, requires pre-training on a source model. Our intention wasn't to imply that LoRA-X uniquely requires pre-training. Rather, the key distinction lies in the type of pre-trained adapters that can be leveraged. ProLoRA offers the significant advantage of being able to transfer a variety of readily available (off-the-shelf) pre-trained adapters—including standard LoRA, DoRA, and FouRA—from a source model to different target models. LoRA-X, in contrast, is restricted to transferring only LoRA-X adapters (when available and requiring fine-tuning on the source model), because it operates solely within the weight subspace, limiting the transfer methodology to that subspace. Therefore, while both require pre-training, ProLoRA's broader compatibility with diverse pre-trained adapters provides greater flexibility and accessibility.
>
> [C5] The full SVD computation has ... benefits over baselines.
>
> [R5] While acknowledging the initial SVD cost, it's a one-time investment for source/target models. ProLoRA amortizes this cost by enabling efficient transfer of diverse pre-trained adapters (style, concepts, LCM) without repeated SVD. Unlike standard LoRA, which requires training adapters from scratch for each task, ProLoRA's one-time SVD is quickly offset by the cumulative savings from multiple adapter transfers.
>
> [C6] It would be great to provide .....  transfer mechanism.
>
> [R6] We appreciate the suggestion for a deeper analysis of subspace/nullspace projection roles. To effectively address this, could you clarify the desired scope? Are you interested in: (1) Layer-wise, (2) Attention-Type (Q, K, V), or (3) Block-Level analysis?
> Also, what type of analysis would be most valuable: (a) Feature Visualization of how each projection affects visual features/styles, or (b) Targeted Ablation Studies isolating the impact of projections in specific layers/attention types?
>
> **Hope responses suffice enough to raise score**

---

> > ### Comment · Reviewer_DiKB · 2025-04-09
> >
> > Thank you to the authors for their response. Most of my concerns have been adequately addressed. Regarding [R6], while I do not suggest a specific experimental design, it would be beneficial to incorporate a broader range of perspectives to more clearly illustrate the distinct roles of subspace and nullspace projection.

---

> > > ### Author Response · Authors · 2025-04-09
> > >
> > > Thanks for the clarification. We would add the analyses in the final version.
> > >
> > > We also add additional experimental results for LoRA-X transfer from SDXL to SSD-1B and compute the CSD-MMD between the adapter when transferred and when trained from scratch.
> > >
> > > | Dataset   | CSD-MMD |
> > > |-----------|---------|
> > > | Bluefire  | 0.0618  |
> > > | Paintings | 0.0391  |
> > > | Origami   | 0.0424  |
> > >
> > > From the results, we see that CSD-MMD is higher compared to ProLoRA as shown in Table 1 (SSD-1B rows). This suggests that our ProLoRA is better at transferring adapters compared to LoRA-X. Hope this addresses your comments fully and these results would be considered in making the final decision.

---

### Decision · Program_Chairs · 2025-05-01

**Decision:**

Accept (poster)

**Comment:**

This paper introduces ProLoRA, a method for zero-shot LoRA adapter transfer using subspace and nullspace projections. Reviewers generally acknowledged the paper's novelty, particularly the nullspace consideration. Its practical utility in reusing adapters without retraining is also valuable.

Several concerns including marginal contribution over LoRA-X, SVD computational cost, unclear baselines, and limited evaluation scope were raised in the first round review. Authors were able to provide detailed replies with additional experiments (including LLMs, LoRA-X comparisons, and new baselines), and clarifications. I think it is a clear acceptance with overall positive scores.